# Immunoproximity biotinylation reveals the axon initial segment proteome

Wei Zhang[1,2,3], Yu Fu[1], Luxin Peng ®[1], Yuki Ogawa ®[3], Xiaoyun Ding ®[3], Anne Rasband[3], Xinyue Zhou[2], Maya Shelly[4], Matthew N. Rasband ®[3] ✉ & Peng Zou ®[1,2,5,6] ✉

The axon initial segment (AIS) is a specialized neuronal compartment required for action potential generation and neuronal polarity. However, understanding the mechanisms regulating AIS structure and function has been hindered by an incomplete knowledge of its molecular composition. Here, using immuno-proximity biotinylation we further define the AIS proteome and its dynamic changes during neuronal maturation. Among the many AIS proteins identified, we show that SCRIB is highly enriched in the AIS both in vitro and in vivo, and exhibits a periodic architecture like the axonal spectrin-based cytoskeleton. We find that ankyrinG interacts with and recruits SCRIB to the AIS. However, loss of SCRIB has no effect on ankyrinG. This powerful and flexible approach further defines the AIS proteome and provides a rich resource to elucidate the mechanisms regulating AIS structure and function.

The axon initial segment (AIS) is the 20–60 μm long proximal region of the axon responsible for action potential generation and maintenance of neuronal polarity[1,2]. Changes in the molecular composition of the AIS, its length, or position can alter neuronal excitability[3–5]. Disruption of the AIS causes axons to acquire dendritic characteristics[6,7]. Recent studies show that AIS disruption occurs in many neurological diseases including autism, Alzheimer's disease, stroke, bipolar disorder, and schizophrenia[8]. Rescuing AIS integrity and function can ameliorate neurological symptoms in Alzheimer's disease and Angelman Syndrome mouse models[9,10]. However, our incomplete knowledge of AIS components hinders our understanding of the structural and functional regulation of AIS in health and disease.

Axon initial segments consist of a specialized extracellular matrix, clustered cell adhesion molecules (CAMs) and voltage-gated ion channels, and a unique cytoskeleton. Among the previously reported AIS proteins, the scaffolding protein ankyrinG (AnkG) is the master organizer for AIS assembly and maintenance[8]. AnkG links membrane proteins to the actin cytoskeleton through a tetramer consisting of βIV and αII-spectrin, and to the microtubule cytoskeleton through the end binding proteins EB1 and EB3[11]. At the AIS, microtubules form parallel fascicles thought to be organized by TRIM46[12,13]. As the site for axonal action potential initiation, AIS also have voltage-gated sodium (e.g., Nav1.2), potassium (e.g., Kv7), and calcium channels (e.g., Cav2) that regulate spike generation, pattern, and shape[14]. NF186, a neuron-specific isoform of the CAM neurofascin (NFASC), assembles and links extracellular matrix molecules (e.g., Brevican) to the AIS cytoskeleton[15]. In addition, some synaptic proteins (e.g., Gephyrin), cisternal organelle molecules (e.g., Synaptopodin) and a Kv1 channel complex can also be found at the AIS[16].

The combination of enzyme-mediated proximity-dependent biotinylation and mass spectrometry-based quantitative proteomics has emerged as a powerful tool to elucidate endogenous protein complexes in subcellular domains. Two main categories of proximity labeling methods have been developed based on enzymes used for catalysis: biotin ligase-based proximity labeling (e.g., BioID and TurboID) and peroxidase-based proximity labeling (e.g., HRP and APEX2).

[1]College of Chemistry and Molecular Engineering, Synthetic and Functional Biomolecules Center, Beijing National Laboratory for Molecular Sciences, Key Laboratory of Bioorganic Chemistry and Molecular Engineering of Ministry of Education, Peking University, Beijing 100871, China. [2]Academy for Advanced Interdisciplinary Studies, PKU-Tsinghua Center for Life Science, Peking University, Beijing 100871, China. [3]Department of Neuroscience, Baylor College of Medicine, Houston, TX, USA. [4]Department of Neurobiology and Behavior, Stony Brook University, New York, NY, USA. [5]PKU-IDG/McGovern Institute for Brain Research, Peking University, Beijing 100871, China. [6]Chinese Institute for Brain Research (CIBR), Beijing 102206, China. ✉e-mail: rasband@bcm.edu; zoupeng@pku.edu.cn

Pioneering experiments using AIS-targeted BioID were recently used to uncover a partial AIS proteome. However, the experiments were restricted to cytoplasmic AIS proteins and the labeling radius of BioID is confined to ~10 nm[17].

Here, we report the development of immunoproximity labeling in fixed neurons, together with a multiple ratiometric analysis strategy to define the AIS proteome (IPL-AIS). This method targets endogenous baits without requiring genetic manipulation, thus avoiding potential artifacts from over-expression or fusion proteins with altered sub-cellular localization[18–20]. This approach also has a larger labeling radius of ~250–500 nm by peroxidase-mediated protein biotinylation in permeabilized and fixed samples[21], allowing us to deeply mine AIS structural components. We applied quantitative ratiometric proteomics with multiple controls to identify AIS proteins[22,23], allowing us to compare the relative expression of proteins in the AIS against other neuronal compartments (axon, dendrites, or soma).

IPL-AIS profiling at days in vitro (DIV) 7, 14, and 21 revealed the dynamic changes in AIS components during neuronal development. Subsequent validation confirmed the identification of previously unreported AIS enriched proteins, including the tumor-suppressor protein scribble (SCRIB encoded by the gene *Scrib*). We found that SCRIB is highly enriched in the AIS in vitro and in vivo. In addition, AnkG interacts with SCRIB and is required for SCRIB localization to the AIS. Together, our experiments define the AIS proteome, which paves the way to understand the molecular mechanisms regulating AIS structure and function.

## Results

### Development of antibody targeted proximity labeling at the AIS

The CAM NF186 is highly and stably localized in the AIS[24,25], therefore we used NF186 for antibody targeted proximity labeling at the AIS. As illustrated in Fig. 1a, cultured cortical neurons were fixed and labeled using antibodies to restrict horseradish peroxidase (HRP) to the AIS. HRP-mediated proximity biotinylation is triggered with the addition of biotin-phenol substrates and hydrogen peroxide. Biotinylation can be evaluated with fluorescent streptavidin and immunoblot analysis. Following enrichment via affinity purification, biotinylated proteins are digested with trypsin and identified via quantitative liquid chromatography-tandem mass spectrometry (LC-MS/MS) analysis.

We optimized antibody dilutions, $H_2O_2$ concentration, reaction time and different biotin-phenol (BP) and biotin-aniline (BA) probes in the proximity labeling system using mature fixed cortical neurons (Supplementary Fig. 1). We found that with optimized conditions we could achieve robust labeling and specificity after just one minute. Previous studies showed the peroxidase substrate BP2 performs well for labeling of cytosolic protein complexes[26], while the substrates BA1 and BA2 exhibit higher reactivity towards nucleic acids (Supplementary Fig. 1e)[27]. Our analysis revealed biotin-phenol (BP) as the most efficient and specific substrate for in vitro AIS proximity labeling. We employed these optimized parameters and found strongly biotinylated proteins localized in the AIS (Fig. 1b). Through imaging, we also tested other AIS-directed antibodies for proximity labeling, including antibodies against AnkG and TRIM46. In our hands, NFASC targeted proximity labeling exhibited the best performance, with less background and highly specific biotinylation of AIS.

To remove background signal arising from non-specific binding of antibodies, experiments omitting the primary antibody (i.e., anti-NFASC) were used as the negative control. To quantify the level of protein enrichment at the AIS, we used NeuN and MAP2 to define soma and somatodendritic domains, respectively. NeuN localizes in the nuclei and cytoplasm of neurons[28,29], while MAP2 is a microtubule-associated protein that is widely distributed throughout the soma and dendrites[5,30]. Fluorescence microscopy showed highly colocalized biotinylation and immunofluorescence labeling for both NeuN and MAP2 (Fig. 1b). Thus, we applied the same experimental workflow to

anti-NeuN and anti-MAP2 defined compartments and used these as references for ratiometric analysis of the AIS proteome.

Biochemical characterization by streptavidin-HRP blot and silver staining showed successful protein biotinylation from anti-NFASC, anti-NeuN, and anti-MAP2 proximity labeling experiments (Fig. 1c). As expected, the negative control sample omitting primary antibody yielded significantly less signal; any background signal presumably arose from detection of endogenously biotinylated proteins. Notably, the biotinylation signal was stronger in anti-NeuN and anti-MAP2 reference samples compared to the anti-NFASC sample, which may reflect the higher abundance of and larger volume occupied by NeuN and MAP2.

To quantitatively compare protein abundance between samples, we designed the following sets of dimethyl labeling-based ratiometric MS proteomic experiments: (1) anti-NFASC vs. no primary antibody; (2) anti-NFASC vs. anti-NeuN; and (3) anti-NFASC vs. anti-MAP2. The first experiment served to remove background labeling arising from non-specific staining of secondary antibody-HRP conjugates, endogenous biotinylated proteins, and non-specific protein adsorption on streptavidin-coated beads. The next two experiments allowed distinction of AIS-specific proteins from those broadly distributed across the cell. For each set of experiments, three biological replicates were performed on cultured cortical neurons at DIV14 (Fig. 1d).

A total of 568 proteins were identified after a stringent cut-off analysis (see "Methods"), among which several known AIS proteins, including TRIM46, AnkG, and NFASC were highly enriched in our dataset (Fig. 1e and Supplementary Data 1). On the other hand, we noticed the low enrichment of EB1 and SEPTIN5, which are also found at the AIS, but not restricted to the AIS[11,17], which explains their lack of specific enrichment in our experiments. Taken together, these results demonstrate that the AIS proteome can be defined using our IPL-AIS methods.

### Identification of AIS proteome at DIV14

Building upon the success of our pilot IPL-AIS experiments, we next sought to improve the quality of our proteomic dataset through implementing the following changes in the workflow (Fig. 2a, b): (1) to improve protein abundance quantitation across multiple samples, we introduced Tandem Mass Tag (TMT) 10-plex isobaric tags to label tryptic peptides[31], (2) to achieve higher coverage in labeled peptides, we reduced sample complexity by fractionating peptides prior to loading onto the LC-MS/MS, and (3) to further improve spatial specificity, we added anti-SMI312 targeted axon proximity labeling to the reference samples (in addition to NeuN and MAP2) (Fig. 2a). Confocal fluorescence imaging analysis confirmed axonal localization of anti-SMI312 targeted protein biotinylation (Supplementary Fig. 2).

Streptavidin-HRP blot analysis confirmed successful enrichment of biotinylated proteins across antibody targeted proximity labeling samples, whereas negative control samples yielded negligible signal (Supplementary Fig. 3). A total of 2755 proteins with at least 2 unique peptides were detected in the 10-plex TMT MS experiment (Fig. 2c). The MS intensity was highly reproducible between replicates (Fig. 2d). We then used the following data analysis pipeline to refine our AIS proteome (Fig. 2c). We first identified biotinylated proteins by comparing the MS intensities of anti-NFASC samples against negative controls (-BP and -1Ab). By applying a cutoff ratio of 2, we obtained a list of 1403 biotinylated proteins. For each protein in this raw list, we calculated the ratios of its MS intensities between anti-NFASC samples versus those in the reference samples (+NeuN, +MAP2, and +SMI312). We then ranked proteins according to their averaged MS intensity ratios (+NF/+NeuN, +NF/ + MAP2, and +NF/ + SMI312) (Fig. 2e) and calculated their averaged rank scores (Fig. 2f). Notably, our results included many known AIS proteins (Figs. 2e, f and 4a), including extracellular proteins (e.g., Brevican and Versican), CAMs (e.g., NFASC and NRCAM), cytoplasmic proteins (e.g., NDEL1 and LIS1), and

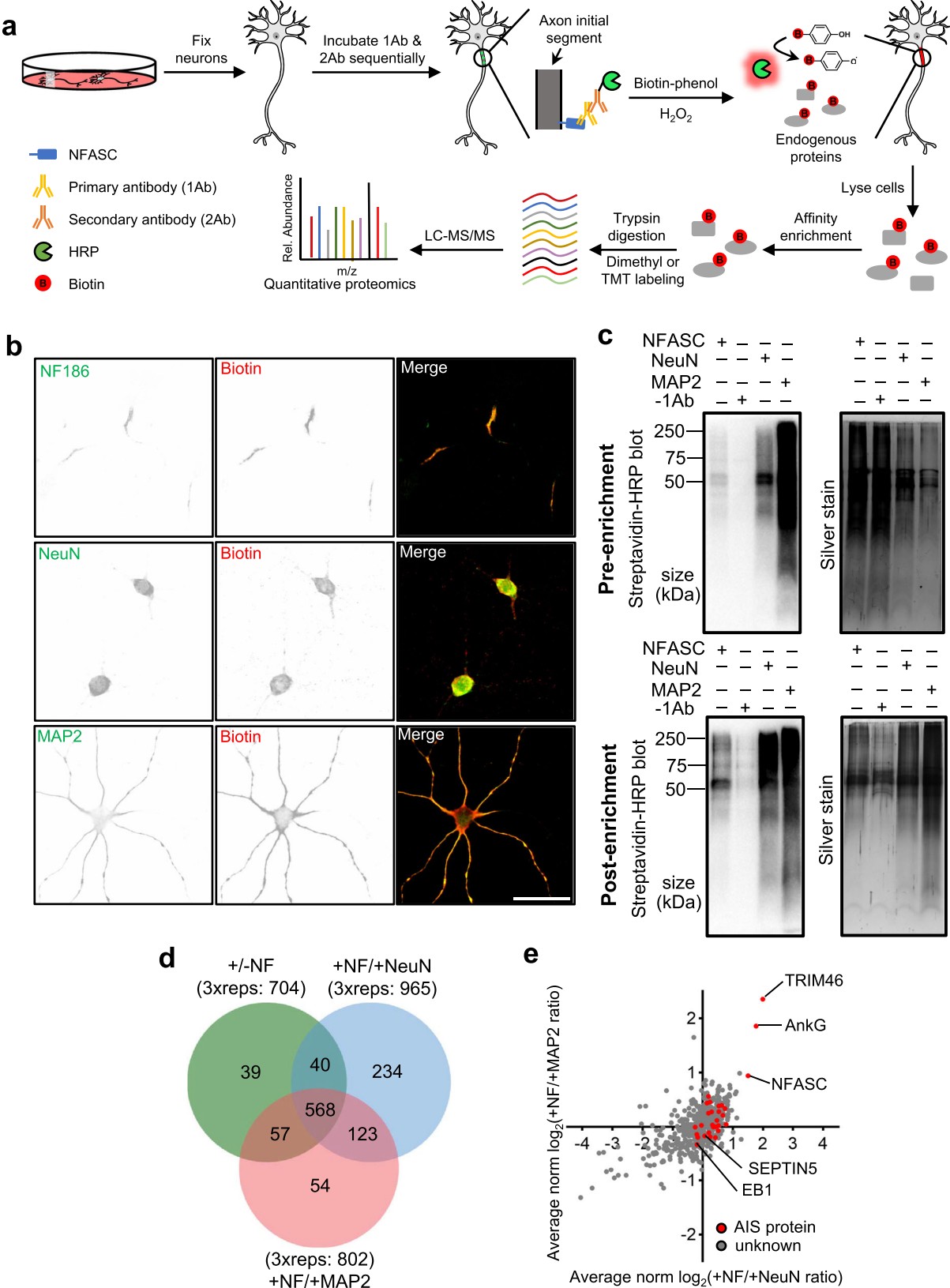

cytoskeletal or cytoskeleton-associated proteins (e.g., βIV-spectrin, αII-spectrin, and TRIM46). This broad coverage of both cytoplasmic and cell surface proteins reflects the large labeling radius of optimized IPL-AIS. Importantly, our dataset included many voltage-gated ion channels, such as sodium channels (e.g., Nav1.2 and Navβ2), potassium channels (e.g., Kv7.2, Kv1.2, Kv2.1, and Kvβ2), and calcium channels

(e.g., Cav2.1 and Cav2.2), consistent with their roles in regulating action potential initiation and shape[14].

AIS candidates (1403) were compared against each reference control and showed their relative amounts in the AIS versus soma, somatodendritic, or axonal domains (Fig. 2e and Supplementary Data 2). AIS highly expressed proteins (NFASC, Nav1.2, AnkG, TRIM46,

**Fig. 1 | Immunoproximity labeling of AIS. a** Experimental scheme of IPL-AIS method. HRP is directed to the axon initial segment in fixed cortical neurons through the specific binding of anti-NFASC antibodies (1Ab) and HRP conjugated secondary antibodies (2Ab). HRP-mediated proximity biotinylation is mediated by the addition of biotin-phenol substrates and hydrogen peroxide. Biotinylated proteins are enriched via affinity purification, digested with trypsin, and identified via LC-MS/MS analysis. **b** Biotinylated proteins colocalize with endogenous protein signals in the axon initial segment, soma, and somatodendrites. The proximity labeling was directed by anti-NFASC, anti-NeuN, and anti-MAP2 antibodies individually in DIV14 cortical neurons with one minute of reaction time. Three independent experiments were performed. Scale bar, 50 μm. **c** Streptavidin blots and silver staining of DIV12 cortical neuron lysates (pre-enrichment) or streptavidin bead eluates (post-enrichment) in conditions of anti-NFASC, no primary antibody (−1Ab), anti-NeuN and anti-MAP2 targeted proximity labeling. Samples of anti-NFASC, -1Ab, anti-NeuN and anti-MAP2 were loaded as the ratio of 4:4:2:1. Two independent experiments were performed. **d** Venn diagrams showing 568 over-lapping biotinylated proteins from +NFASC/+NeuN, +NFASC/+MAP2, and filtered +/-NFASC datasheet. Three independent dimethyl proteome replicates were performed at DIV14 for each condition. **e** Scatterplot showing the enrichment of anti-NFASC captured biotinylated proteins over soma (x-axis) and somatodendrites (y-axis) intensity in the reference of average normalized $\log_2$(H/L ratios). Red dots indicate reported AIS components and gray dots are proteins having no AIS information.

and βIV-spectrin) were always ranked at the top, while more widely expressed AIS proteins differentiated according to their relative expression level in subcellular domains, such as the microtubule protein TUBA4A or microtubule related protein MAP6 (Fig. 2e)[17]. To better define AIS enriched proteins, we applied averaged rank scores to re-list the known and putative AIS proteins (Fig. 2f and Supplementary Data 2). Consistently, highly expressed, known AIS proteins were found among the top 5. More widely expressed proteins, such as F-actin monooxygenase MICAL3 (ranked 371) and Rho GTPase activating protein ARHGAP21 (ranked 455), were not as highly ranked[17].

To obtain a high-confidence AIS proteome list, we took the overlap of the top 200 proteins in each list of MS intensity ratios (+NF/+NeuN, +NF/+MAP2, and +NF/+SMI312). The resulting high-confidence list contained 71 proteins (Fig. 2c, g and Supplementary Data 2). Gene Ontology cellular compartment analysis revealed an overrepresentation of AIS and node of Ranvier terms due to the similar compositions of these two compartments (Fig. 2h). The remaining top 8 GOCC terms were consistent with AIS characteristics and its endocytosis functions[16,32].

## Mapping the AIS proteome across neuronal maturation

The AIS is developmentally and physiologically dynamic[25,33]. To determine dynamic developmental changes in the AIS proteome, we performed IPL-AIS at DIV7, DIV14, and DIV21. Two parallel 10-plex TMT experiments were performed and DIV14 samples were used as a reference to bridge DIV7 and DIV21 samples (Fig. 3a).

We first analyzed the DIV7 AIS proteome. Parallel analysis of 1532 common proteins revealed highly reproducible protein quantifications ($R^2 = 0.94$) (Fig. 3b). After stringent filtering (Supplementary Fig. 4a), we obtained 1407 biotinylated proteins. Rank analysis revealed the specificity of our DIV7 AIS dataset with top-ranked AIS proteins including AnkG, TRIM46, βIV-spectrin, NFASC, and Nav1.2 (Fig. 3c). Other AIS proteins were more widely dispersed, possibly due to different expression levels and labeling efficiencies. Through integrating average rank scores of AIS versus soma or somatodendrites, we obtained a final ranked putative AIS proteome including AIS proteins in top positions (Fig. 3d and Supplementary Data 3). Using the same workflow (Supplementary Fig. 4b), we identified 1738 biotinylated proteins at DIV21. Parallel analysis of anti-NFASC replicates showed highly reproducible protein quantifications ($R^2 = 0.98$) (Fig. 3e), and rank scores of AIS versus soma or somatodendritic proteins (Fig. 3f) resulted in AIS proteins being consistently at the top (Fig. 3g and Supplementary Data 3). Some AIS proteins, such as Synaptopodin (SYNPO) and PSD-93 (PSD93) were ranked lower (Fig. 3g)[34,35].

Comparing biotinylated proteins between DIV7 and DIV14 in 10-plex TMT revealed 173 proteins only present at DIV7 and 152 proteins only present at DIV14, among which there were only two previously reported AIS proteins: Gephyrin and microtubule protein TUBA4A (Supplementary Fig. 4d). DIV14 samples in the two parallel 10-plex TMT experiments presented a very high reproducibility among biological replicates (Supplementary Fig. 4e) and enrichment of known AIS proteins (Supplementary Fig. 4f), allowing us to compare relative protein expression in three stages together. To identify the core set of putative AIS proteins during development, DIV7, 14, 21 and the prior DIV14 datasets (Fig. 2) were combined, which generated a total of 549 common proteins (Fig. 3h). Among these, 534 proteins showed 20% changes in any one pair comparison of the three time points and were used for heatmap clustering. Six clusters were generated with distinct expression patterns (Fig. 3i and Supplementary Data 3). 51.7% of proteins were gradually upregulated and 4.1% of proteins were gradually downregulated along neuronal development (Fig. 3j). The cluster with the highest expression level at DIV21 included 83.5% of proteins, while at DIV7 the cluster with the highest expression level included only 15.3% of proteins.

Consistently, the previously reported AIS proteins exhibited the highest expression level at DIV21, including NFASC (*Nfasc*), AnkG (*Ank3*), Nav1.2 (*Scn2a*), and βIV-spectrin (*Sptbn4*), while TRIM46 (*Trim46*) decreased along neuronal development (Fig. 3k). Quantitative analysis revealed the fold changes and statistical significance between stages (Supplementary Data 3). For example, microtubule-associated proteins MAP1A (*Map1a*), MAP6 (*Map6*), sodium/potassium-transporting ATPase subunit alpha-1 (*Atp1a1*) and casein kinase II subunit alpha (*Csnk2a1*) exhibited a more than two-fold significant change from DIV7 to 14 (Fig. 3l). From DIV14 to 21, more microtubule or microtubule-associated proteins like TUBB5 (*Tubb5*), TUBB3 (*Tubb3*), and EB3 (*Mapre3*) significantly increased, while calcium/calmodulin-dependent protein kinase KCC2D (*Camk2d*) and KCC2A (*Camk2a*) also had more than two-fold changes (Fig. 3m). Comparison between DIV7 and 21 revealed more components with a two-fold significant increase, including the AIS proteins NFASC (*Nfasc*), Nav1.2 (*Scn2a*), and GABA(A) receptor subunit gamma-2 (*Gabrg2*), while TRIM46 (*Trim46*) showed a two-fold significant decrease (Fig. 3n), further demonstrating the dynamic nature of AIS composition along neuronal development.

## SCRIB is a bona fide AIS enriched protein

To identify previously unreported AIS proteins, we focused on genes highly ranked in our DIV14 dataset but not previously reported at the AIS (Fig. 4a). Three of the top-ranked hits *Wdr7*, *Scrib*, and *Wdr47* were selected for further analysis and to assess their subcellular localization in neurons. WD repeat-containing protein 7 (WDR7) mediates V-ATPase-dependent vesicle acidification in kidney cell lines and neuroendocrine cells[36,37]. In neurons WDR7 may regulate synaptic vesicle acidification[38]. WDR47 is required for neuronal polarization and axonal and dendritic development[39,40]. The scaffold protein SCRIB has been reported as an important regulator for apical dendrite development, spine morphology, and synapse plasticity[41–43].

We first leveraged CRISPR/Cas9-based homology-independent genome editing to integrate spaghetti monster fluorescent protein with V5 tags (smFP-V5) into *Wdr7*, *Scrib*, and *Wdr47* to create C-terminal fusion proteins[44]. Cultured neurons were infected at DIV0 by AAV delivery and observed at DIV14. We found SCRIB was highly and specifically enriched at the AIS, while WDR47 and WDR7 were present at the AIS, but not specifically enriched there (Fig. 4b). Quantification of the smFP-V5 signal in the AIS reveals SCRIB is 17 ± 2.55

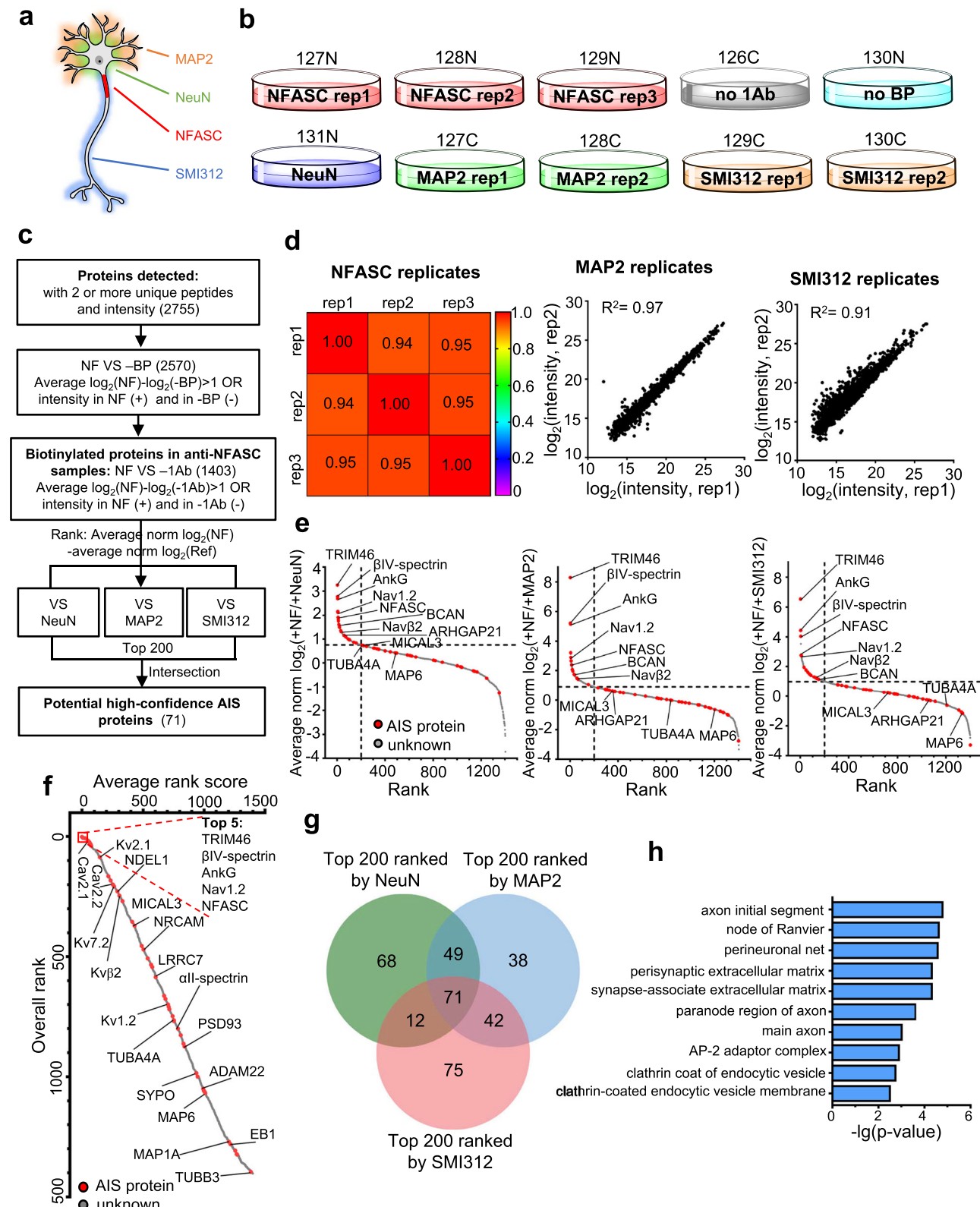

times higher in the AIS than in proximal dendrites, while WDR47 and WDR7 are $2.21 \pm 0.42$ and $1.12 \pm 0.12$ fold higher in the AIS than proximal dendrites, respectively (Fig. 4c).

To further investigate AIS SCRIB, we treated neurons after endogenous tagging (smFP-V5) of SCRIB using 0.5% Triton X-100 to reveal the detergent-resistant pool of SCRIB (Fig. 5a). The detergent insoluble SCRIB was retained and highly enriched at the AIS where it colocalized with the AIS cytoskeletal protein βIV-spectrin; these

observations are consistent with previous studies showing that AIS proteins are resistant to detergent extraction[45,46]. Some AIS proteins, including voltage-gated sodium channels, βIV-spectrin, NFASC, and AnkG have a periodic spacing in the AIS of approximately 180–190 nm[47,48]. To test whether SCRIB also exhibits this pattern, we used stimulated emission depletion (STED) nanoscopy to image smFP-V5-tagged SCRIB. We found that AIS SCRIB has a periodicity of $189.4 \pm 6.6$ nm in DIV16 neurons (Fig. 5b). Together, these results

**Fig. 2 | The axon initial segment proteome at DIV14. a** Schematic of neurons biotinylated at the axon initial segment by targeting NFASC (red), soma by targeting NeuN (green), somatodendrites by targeting MAP2 (orange), and axon by targeting phosphorylated neurofilaments using SMI312 antibody (blue). **b** Design of 10-plex TMT for AIS quantitative proteomic experiments. Samples were collected from at least three independent primary neuron cultures for each condition. TMT tags were labeled on the dishes. **c** Workflow for AIS proteome analysis. **d** Correlation analysis of biological replicates showing high reproducibility. Pearson correlation was applied to parallel analysis of 2377, 2691, and 2498 common elements in anti-NFASC, anti-MAP2, and anti-SMI312 replicates, respectively. **e** Ranking 1403 biotinylated proteins based on their averaged MS intensity ratios. Red dots indicate previously reported AIS components and gray dots are proteins

having no AIS information. **f** Overall rank plot integrating three reference proteomes (**e**) listing potential AIS enriched proteins in the top positions. Proteins are sorted by average rank scores from +NFASC/+NeuN, +NFASC/ + MAP2, and +NFASC/ + SMI312 experiments. Red dots indicate reported AIS components and gray dots are proteins having no AIS information. Some AIS proteins are labeled. **g** Venn diagrams showing 71 high-confidence AIS proteins from the top 200 candidates from +NFASC/+NeuN, +NFASC/ + MAP2, and +NFASC/ + SMI312 proteomes. **h** Top 10 cellular compartment terms. 71 high-confidence AIS candidates were applied to Gene Ontology cellular compartment analysis by PANTHER overrepresentation with Fisher's exact test. A total of 2744 valid proteins in this TMT were used as a reference (2755 proteins identified and 2744 valid data in Gene Ontology).

suggest a strong association of SCRIB with previously described components of the AIS periodic cytoskeleton[47].

To determine whether SCRIB localizes to the AIS in vivo, we performed intraventricular injection in Cas9 transgenic mice using AAV to express gRNA to target endogenous mouse *Scrib* and introduce smFP-V5 into the c-terminus of SCRIB. We confirmed the gRNA targeting mouse *Scrib* resulted in detection of AIS SCRIB in cultured mouse hippocampal neurons (Supplementary Fig. 5a). Three weeks after injection of AAV into P0 Cas9 mouse pups, we found strong V5 immunolabeling of AIS that colocalized with βIV-spectrin (Fig. 5c). Thus, SCRIB is present at the AIS both in vitro and in vivo.

To further confirm SCRIB localization at the AIS, we used commercial antibodies for SCRIB labeling. We used 2 different antibodies targeting different antigens of human SCRIB (amino acids 1100–1400 and 1568–1630; see "Methods"). Consistent with our *Scrib* knock-in results, antibody labeling also showed strong and specific SCRIB enrichment at the AIS with and without detergent extraction (Fig. 6a and Supplementary Fig. 5b). We validated the specificity of the immunostaining using CRISPR/Cas9 mediated *Scrib* knock-out. We infected cultured neurons using AAV to express an HA-tag and three gRNA targeting *Scrib*, or AAV expressing HA and template; transduction was performed at DIV0 and neurons were fixed 14 days later. SCRIB was still enriched in the AIS after transduction with AAV expressing the template, but AIS labeling was lost in neurons transduced with the *Scrib* gRNAs (Fig. 6b). We generated two different AAVs with triple gRNA each targeting *Scrib*. Quantification showed a ~75% and 82% reduction in AIS SCRIB positive neurons after transduction with these AAVs (SCRIB positive AIS: 88.9% ± 1.5% for control, 13.7% ± 6.8% for *Scrib* gRNA1, and 6.7% ± 2.7% for *Scrib* gRNA2; Mean ± SEM, one-way ANOVA) (Fig. 6c). In addition, loss of SCRIB did not affect AnkG clustering. We found AIS AnkG was comparable in the presence or absence of SCRIB (Fig. 6b, d). Finally, using the validated antibodies, we found SCRIB is enriched in the AIS of neurons in cerebral cortex (Fig. 6e). These results show that SCRIB is an AIS protein.

## AnkG recruits SCRIB to the AIS

To determine when SCRIB becomes clustered at the AIS, we analyzed its expression and distribution along neuronal development. At DIV3 we found AnkG was highly enriched in the proximal axon, while SCRIB was not (Fig. 7a). By DIV7 we found clustered and enriched SCRIB that colocalized with AnkG at the AIS (Fig. 7b). Quantification of SCRIB and AnkG enrichment throughout development showed that AnkG was enriched in the proximal axon in 52.8% ± 10.3% neurons at DIV3 and in more than 90% neurons after DIV7 (Fig. 7c). In contrast, AIS SCRIB was present in only 15.1% ± 5.8% of neurons at DIV3 and increased to 67.8% ± 7.4%, 75.8% ± 4.4%, and 93.5% ± 2.0% at DIV7, 14, and 21, respectively (Fig. 7c). These results suggest that SCRIB clustering at the AIS follows AnkG clustering.

Previous studies show that AnkG is the earliest AIS protein and is required for AIS formation and maintenance[8,16]. To determine if AnkG also contributes to AIS clustering of SCRIB, we used CRISPR/

Cas9 mediated knockout of AnkG. Neurons were infected with AAV expressing template or *Ank3* triple gRNA at DIV0 and then fixed at DIV14. In control conditions SCRIB colocalized with AnkG. However, AIS SCRIB was not detected in AnkG deficient neurons (Fig. 7d). We found a ~68% decrease of AIS AnkG positive neurons in *Ank3* gRNA conditions (97.5% ± 1.4% for control; 29.4% ± 5.0% for *Ank3* gRNA conditions; Mean ± SEM, p = 0.0002, unpaired t test) (Fig. 7e). Similarly, neurons with AIS SCRIB also showed a ~63% decrease after loss of AnkG (94.5% ± 1.7% for control; 31.7% ± 5.8% for *Ank3* gRNA conditions; Mean ± SEM, p = 0.0005, unpaired t test) (Fig. 7f). These observations suggest that AnkG is necessary for SCRIB clustering at the AIS.

To further define the relationship between AnkG and SCRIB, we performed co-immunoprecipitation (IP) experiments using HEK293T cells transfected with flag-tagged SCRIB (Scrib-Flag) and GFP-tagged AnkG (AnkG270-EGFP) or pEGFP-N1. Using anti-Flag antibodies, we successfully co-immunoprecipitated AnkG270-EGFP (Fig. 7g). Similarly, Scrib-Flag co-immunoprecipitated with AnkG-EGFP when the latter was immunoprecipitated using anti-GFP antibodies (Fig. 7h). Together, these results suggest that SCRIB interacts directly with AnkG.

To further define how SCRIB interacts with AnkG, we used various *Scrib* truncated constructs omitting N-terminal 16 leucine-rich repeats (ΔLRR), 4 PDZ domains (ΔPDZ), or spacer regions between the last LRR and the first PDZ (ΔIMR) (Fig. 7i); the SCRIB was also fused to dTomato. AnkG270-EGFP and truncated *Scrib* were co-transfected in HEK293T cells and AnkG270-EGFP was immunoprecipitated. We found that SCRIB lacking its IMR amino acids failed to interact with AnkG (Fig. 7j). In addition, compared with dTomato-Scrib-ΔPDZ, dTomato-Scrib-ΔLRR had lower binding to AnkG (Fig. 7j), suggesting that the N-terminus of SCRIB is critical for AnkG binding.

## Discussion

The AIS plays key roles in regulating action potential initiation and maintenance of neuronal polarity. Functional and structural changes to the AIS in normal and pathological conditions are often associated with changes in its molecular composition. Here, we used IPL-AIS to identify the AIS proteome. Our experiments identified nearly all previously reported AIS components including extracellular matrix proteins, membrane proteins, and cytoskeleton-associated proteins. Notably, most voltage-gated ion channels and their accessory subunits were also identified, including Nav1.2, Navβ2, Kv2.1, Kv7.2, Kv1.2, Kvβ2, and Cav2.1 and Cav2.2 (Fig. 2f). The results reported here complement and extend a previous proximity labeling study using BioID[17] especially as it relates to cell surface proteins, since the prior study was biased towards cytoplasmic proteins.

Although we used stringent thresholding, there are likely non-AIS proteins in our dataset owing to non-specific absorption of antibodies, signal amplification after biotinylation, and the large labeling radius of the horse-radish peroxidase. Nevertheless, our ratiometric analysis against multiple controls allowed for the strong enrichment of AIS proteins including AnkG, βIV-spectrin, NFASC,

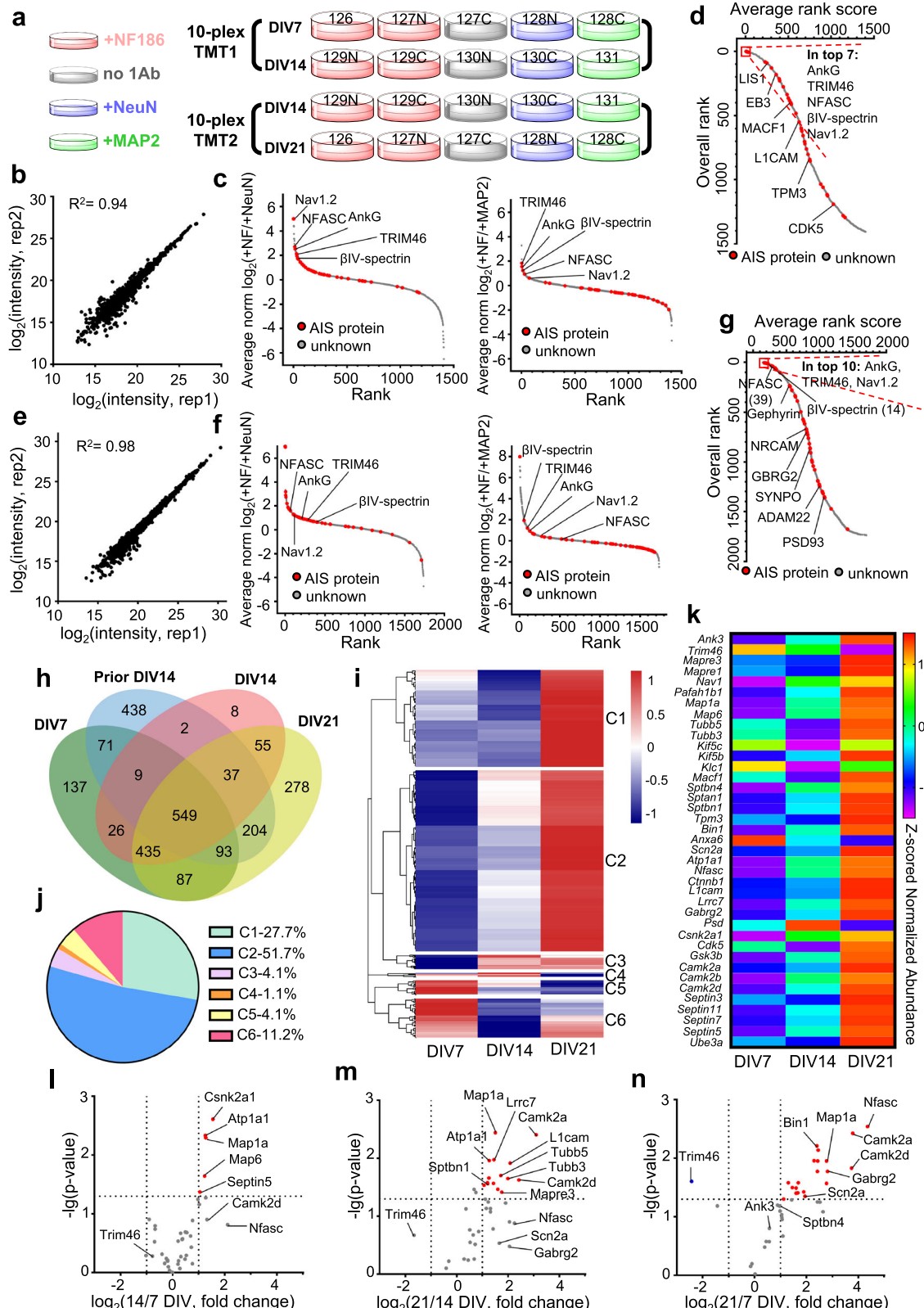

Nav1.2 and TRIM46 among others (Figs. 2 and 3). The dispersed distribution of some other known AIS proteins might be attributed to their wide distribution in neurons, varied protein abundance at the AIS, or differences in labeling efficiency. In addition, our control experiments may also be useful in future studies to examine the relative expression of proteins in different neuronal compartments (AIS, axon, dendrites, and soma).

The AIS composition may change during neuronal development and AIS maturation. For example, recent studies revealed that NuMA1 and the P2Y1 purinergic receptor participate only in AIS assembly[25,49]. We applied TMT-based quantitative mass spectrometry to identify developmental changes in the AIS proteome, revealing a gradual increase of most AIS proteins, including the known AIS proteins βIV-spectrin, AnkG, NFASC and voltage-gated ion channels; these

**Fig. 3 | The axon initial segment proteome across development. a** Experimental design to study the AIS proteome during development using two parallel 10-plex TMT. **b** Pearson correlation showing high reproducibility in anti-NFASC targeted AIS proximity labeling for DIV7 cortical neurons (1532 common elements from 2 replicates). **c** Ranking DIV7 AIS biotinylated 1407 proteins according to their averaged MS intensity ratios (+ NFASC/+NeuN and +NFASC/ + MAP2). **d** DIV7 overall rank plot listing potential AIS proteins. **e** Pearson correlation showing high reproducibility in anti-NFASC targeted AIS proximity labeling for DIV21 cortical neurons (1859 common elements from 2 replicates). **f** Ranking DIV21 AIS biotinylated 1738 proteins according to their averaged MS intensity ratios (+ NFASC/ +NeuN and +NFASC/ + MAP2). **g** DIV21 overall rank plot listing potential AIS proteins. **h** Venn diagrams showing 549 common AIS candidates at DIV7, 14 and 21. The prior AIS proteome at DIV14 was also included in this analysis. **i** Six clusters with distinct protein expression profiles were generated by pheatmap analysis along neuronal development. **j** The percentage of protein numbers in each cluster among 534 analyzed proteins. **k** AIS protein expression patterns along neuronal development. Proteins are represented by their gene names. **l–n** Volcano plots showing fold changes in AIS proteins between DIV7 and 14 (**l**), DIV 14 and 21 (**m**), or DIV7 and 21 (**n**). Red and blue points represent AIS proteins with significant up-regulation and down-regulation, respectively. Proteins are represented by their gene names. Horizontal dashed lines indicate $p = 0.05$, and unpaired two-tailed t test was used for $p$-value calculation. Vertical dashed lines indicate the cutoff of $\log_2$(fold change) = ±1.

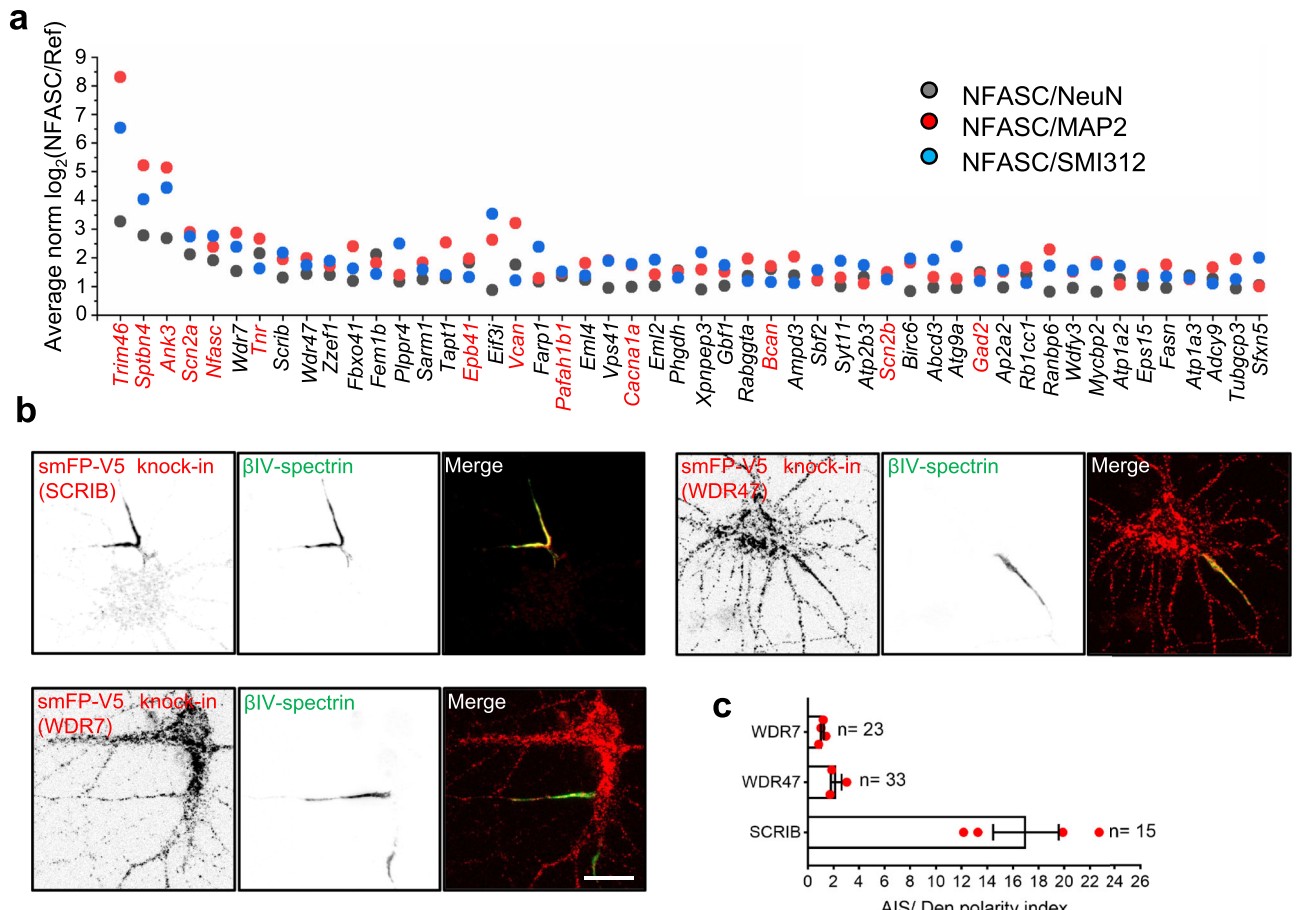

**Fig. 4 | Screening AIS candidates by tagging endogenous genes. a** Top 50 ranked AIS candidates at DIV14. The fold change of proteins intensity in the AIS compared with soma, somatodendrites, and the axon. Previously reported AIS proteins are shown in red, and the rest are shown in black. Proteins are represented by their gene names. **b** Representative images of smFP-V5 tag knock-in neurons for SCRIB, WDR47, and WDR7. Neurons were infected with two AAVs, one is for Cas9 expression and another for gRNA and donor expression for homology-independent knock-in at DIV0. Samples were fixed at DIV14 and labeled with βIV-spectrin (green, AIS) and V5 tag (red, targeted proteins). Scale bar, 20 μm. **c** Quantification of V5 tag mean intensity in the AIS versus in proximal dendrites from knock-in samples. Four independent experiments were performed for WDR7 ($n = 23$ neurons) and SCRIB ($n = 15$ neurons). Three independent experiments were performed for WDR47 ($n = 33$ neurons). Data are shown as mean ± SEM.

observations are consistent with previous analyses of AIS development using immunostaining methods[33]. While as many as 83.5% of proteins identified exhibited the highest level of enrichment at DIV21, we observed some proteins that deviated from this general trend. For example, enrichment of the AIS protein TRIM46 reached its maximum at DIV7. This observation is in accordance with the proposed role of TRIM46 on establishing early neuronal polarity and axon specification[13].

As an example of the utility of our approach to identify new AIS proteins, we examined more closely the top three candidates that were not previously reported at the AIS. Among them, we found SCRIB

(ranked 8) is highly and specifically enriched at the AIS, although all three showed some degree of enrichment. We further confirmed SCRIB's AIS localization by (1) smFP-V5 tagging of endogenous SCRIB using CRISPR, (2) immunostaining, (3) knock-out, (4) detergent resistance, (5) periodicity like that of other AIS proteins, and (6) binding to and dependence on AnkG. In the future we will perform similar analyses on other candidates that are highly enriched in the AIS proteome reported here.

SCRIB was previously reported to regulate neuronal migration, apical dendrite development, axonal connectivity and synaptic plasticity[41,43,50]. *SCRIB* deletion in humans causes severe dysmorphic

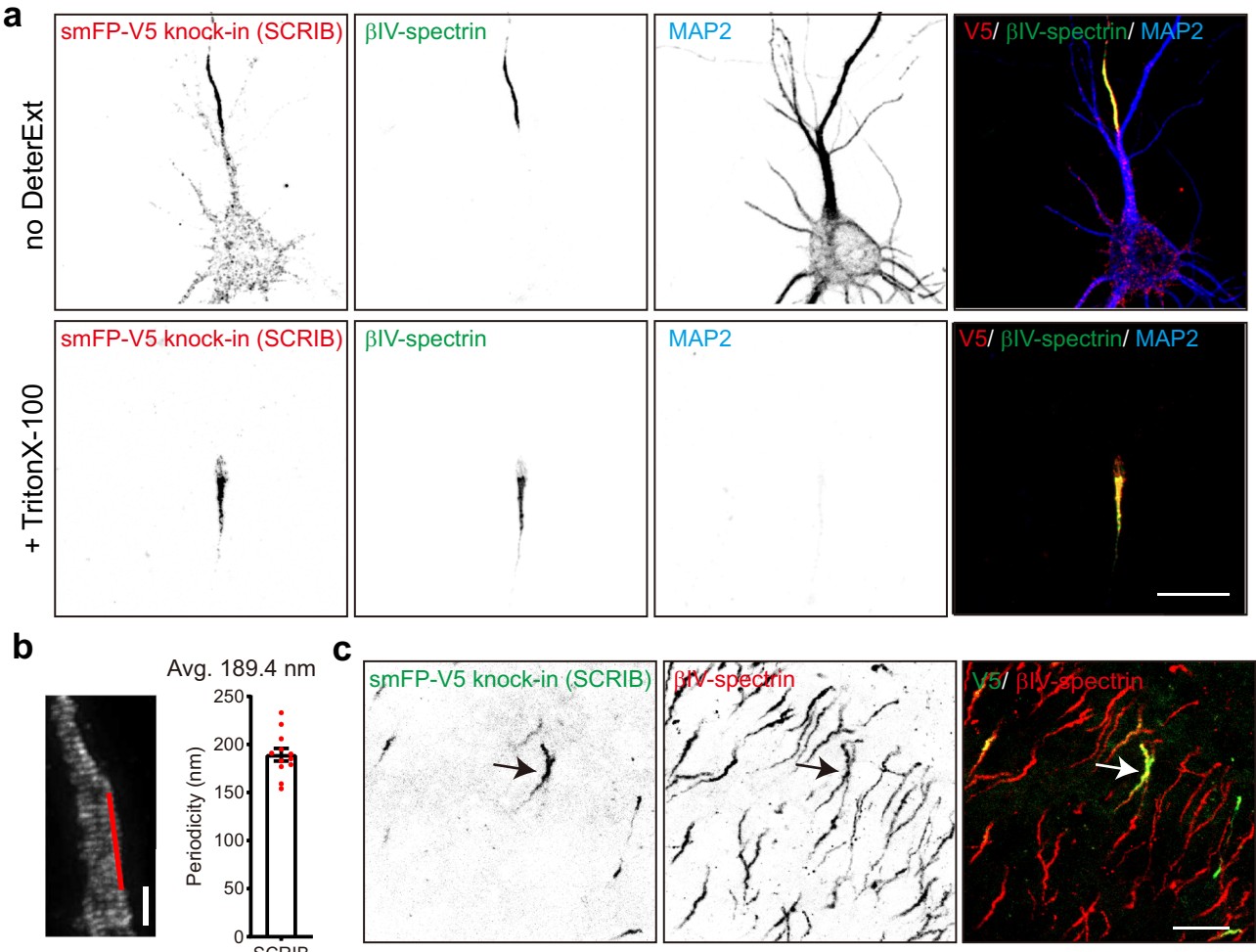

**Fig. 5 | Validation of SCRIB enrichment at the AIS in vitro and in vivo by knock-in. a** Representative images of smFP-V5 tag knock-in to endogenous SCRIB. Live neurons were treated with or without 0.5% Triton X-100. Neurons were infected at DIV0. After 14 days, samples were immunolabeled for the V5 tag (red, endogenous SCRIB), βIV-spectrin (green, AIS), and MAP2 (blue, somatodendrites). Three independent experiments were performed. Scale bar, 20 μm. **b** A representative STED image of smFP-V5 tagged SCRIB. Knock-in neurons were fixed at DIV16, and stained with V5 tag antibody. Regions indicated by red lines were used to generate the intensity profiles, and the average periodicities were calculated. $n = 12$ AIS from one experiment were used for quantification. Data is mean ± SEM. Scale bar, 1 μm. **c** Representative image of smFP-V5 tagged SCRIB in the cortex. P0 Cas9 pups were intraventricularly injected with mouse-specific *Scrib* gRNA and donor AAV and sacrificed at postnatal day 23 (P23). Four mice were used to repeat this experiment. Samples were stained for the V5 tag (green, endogenous SCRIB) and βIV-spectrin (red, AIS). Scale bar, 20 μm.

features[51], and other pathogenic variants cause neural tube defects and craniorachischisis[52,53]. Similarly, SCRIB mutants lacking the last two PDZ and the C-terminal domains die before or at birth with severe brain malformation[41]. Heterozygous or conditional knockout mice also show psychomotor deficits and autism-like behaviors[41,50]. However, no report has described SCRIB at the AIS. We found that SCRIB's AIS localization is AnkG-dependent. Although loss of SCRIB did not affect AnkG, altered AnkG expression is associated with epilepsy, and psychiatric disorders, including bipolar spectrum disorder and schizophrenia[8]. Since AnkG maintains SCRIB at the AIS, we speculate that loss of SCRIB function might be a core molecular pathology of AnkG-related neurodevelopmental disorders. Exploring SCRIB's function at the AIS both physiologically and pathologically will be of considerable interest in future studies.

In conclusion, the results reported here illustrate the power of IPL-AIS to define molecular complexes in distinct subcellular domains. Furthermore, the results reported here are a rich resource to identify other AIS-enriched proteins that may contribute to AIS structure and function.

## Methods

### Animals

P0 Sprague-Dawley rat pups were purchased from Peking University Health Science Center. E18.5 timed-pregnant Sprague-Dawley rats were purchased from Charles River Laboratories. Rat pups and embryos were used for the neuron culture. P7 wild-type C57BL/6 mice were purchased from Beijing Vital River Laboratory Animal Technology Co., Ltd. Transgenic Cas9 mice (JAX stock #027650) were obtained from The Jackson Laboratory. P0 Cas9 pups were used for neuron culture and intraventricular injection to tag endogenous SCRIB in vitro and in vivo. Animals were maintained in a facility with 22 °C temperature, 40–60% humidity and a normal 12 h light/dark schedule. Animals were free to have food and water. All experimental procedures were performed in accordance with the guidelines of the Institutional Animal Care and Use Committees of Peking University and Baylor College of Medicine (IACUC #AN-4634).

## Probe synthesis

**Synthesis of compound 1.** To a solution of D-biotin (1.76 g, 7.2 mmol) and EDCI (1.54 g, 8.0 mmol) in 50 ml DMF, N-hydroxysuccinimide (0.92 g, 8.0 mmol) was added at room temperature (RT). The reaction mixture was stirred at 50 °C overnight. DMF was removed under vacuum to a residue of 5 ml. Add 100 ml cold ethanol into the residue during which time a white precipitate formed. Precipitate was filtered and washed with 15 ml ethanol twice, then dried in vacuum to afford compound **1**. The yield was 73%.

$^1$H-NMR (400 MHz, $d_6$-DMSO): 6.43 (1H, s), 6.37 (1H, s), 4.30 (1H, m), 4.15 (1H, m), 3.11 (2H, m), 2.84 (1H, dd), 2.81 (4H, s), 2.67 (2H, t), 2.60 (1H, d), 1.75–1.30 (6H, m).

**Synthesis of compound 2.** Trimethylamine (1.60 g, 16 mmol) was added to a solution of mono-Boc protected ethylene diamine (1.66 g, 10 mmol) and compound **1** (1.76 g, 5.2 mmol) in 50 ml DMF. The reaction mixture was stirred at RT overnight and concentrated by rotary evaporation. The residue was dissolved in a solution of DCM and $^i$PrOH (v/v = 4:1) and washed with 20 ml 1 M HCl followed by 20 ml water three times. The organic layers were combined and evaporated, and the residue was purified by column chromatography, affording compound **2**. The yield was 72%.

$^1$H-NMR (400 MHz, $d_6$-DMSO): 7.79 (1H, t), 6.79 (1H, t), 6.43 (1H, s), 6.36 (1H, s), 4.30 (1H, m), 4.13 (1H, m), 3.05 (3H, m), 2.96 (2H, m), 2.81 (1H, dd), 2.59 (1H, d), 2.04 (2H, t), 1.68–1.41 (4H, m), 1.37 (9H, s), 1.34–1.19 (2H, m).

**Synthesis of BP and BA1.** To a solution of compound **1** (0.50 g, 1 e.q.) and corresponding primary amine (1.1 e.q.) in 50 ml DMF, trimethylamine (3 e.q.) was added to the mixture and stirred overnight at RT. The reaction solvent was evaporated and the residue was purified by C18 reverse phase column (Waters XBridge Prep C18 5 μm OBD 19 × 150 mm) on semi-preparative UPLC (Waters 2998 Photodiode Array Detector and 2545 Binary Gradient Module) with a gradient of 3% to 60% methanol in water over 25 min. The overall yields were 60–70%.

$^1$H-NMR for BP (400 MHz, $d_6$-DMSO): 9.13 (1H, s), 7.79 (1H, t), 6.97 (2H, d), 6.66 (2H, d), 6.42 (1H, s), 6.35 (1H, s), 4.31 (1H, m), 4.12 (1H, m), 3.18 (2H, dd), 3.08 (1H, m), 2.83 (1H, dd), 2.57 (3H, m), 2.03 (2H, t), 1.65–1.39 (4H, m), 1.35–1.19 (2H, m).

$^1$H-NMR for BA1 (400 MHz, $d_6$-DMSO):7.78 (1H, t), 6.84 (2H, d), 6.49 (2H, d), 6.43 (1H, s), 6.36 (1H, s), 4.85 (2H, s), 4.31 (1H, m), 4.12 (1H, m), 3.16 (2H, dd), 3.09 (2H, dd), 2.84 (1H, dd), 2.56 (3H, m), 2.03 (2H, t), 1.38–1.66 (4H, m), 1.28 (2H, m).

**Synthesis of BP2 and BP3.** To a solution of D-biotin (1.00 g, 4.0 mmol), EDCI (0.85 g, 4.5 mmol) in 30 ml DMF, corresponding phenol or aniline (0.56 g, 4.5 mmol) was added and the reaction mixture was stirred at RT overnight. The reaction mixture was evaporated, and the residue was purified by column chromatography, affording compound **L1**.

To the solution of **L1** in 20 ml DCM, BBr$_3$ (1.0 ml, 10 mmol) was added to the mixture dropwise at 0 °C. After being stirred at 0 °C for 4 h, the mixture was heated to RT and continued to react for 12 h. Then, 50 ml water was added to the mixture, and the residue was washed by 10 ml cold water for three times, and purified by semi-preparative column to get the purified product BP2 or BP3. The whole yield was 25%.

$^1$H-NMR for BP2 (400 MHz, $d_6$-DMSO): 9.60 (s, 1H), 9.13 (s, 1H), 7.35 (d, 2H), 6.67 (d, 2H), 6.45 (s, 1H), 6.37 (s, 1H), 4.31 (m, 1H), 4.14 (m, 1H), 3.14 (m, 1H), 2.84 (dd, 1H), 2.59 (d, 1H), 2.25 (t, 2H), 1.69–1.46 (m, 4H), 1.42–1.27 (m, 2H).

$^1$H-NMR for BP3 (400 MHz, $d_6$-DMSO): 9.44 (s, 1H), 6.90 (m, 2H), 6.76–6.74 (m, 2H), 6.47 (s, 1H), 6.38 (s, 1H), 4.31 (m, 1H), 4.15 (m, 1H), 3.13 (m, 1H), 2.83 (dd, 1H), 2.58 (d, 1H), 2.53 (m, 2H), 1.71–1.47 (m, 4H), 1.45–1.34 (m, 2H).

**Synthesis of BP4.** Compound **2** (0.45 g, 1.2 mmol) was dissolved in 15 ml TFA/DCM (v/v = 1:1) and stirred at RT for 2 h. The solvent was removed under vacuum and the residue was dissolved in 20 ml DMF.

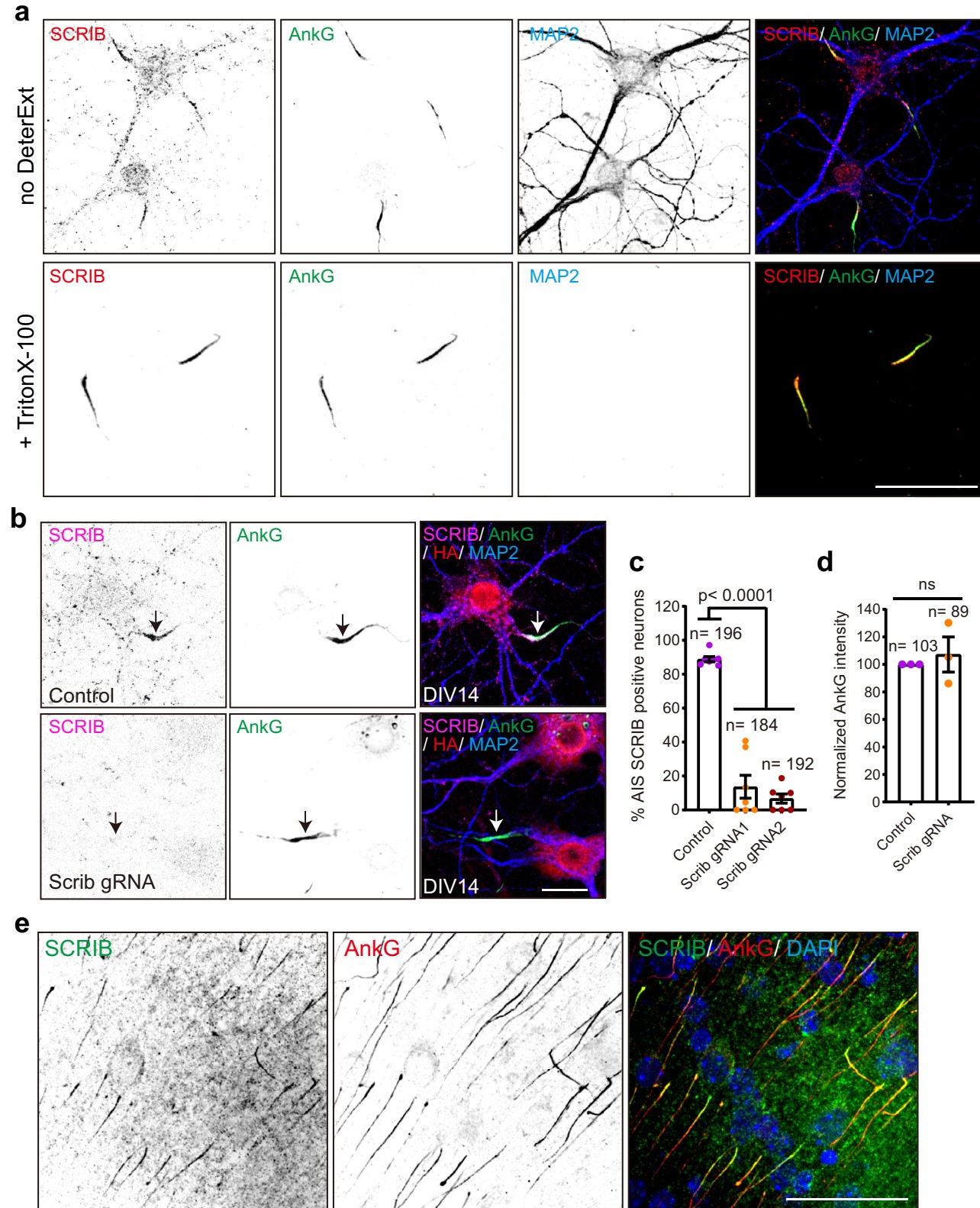

To this solution was added 4-Hydroxybenzoic acid (0.15 g, 1.2 mmol), EDCI (0.45 g, 2.4 mmol), HOBt (0.35 g, 2.4 mmol), and TEA (0.36 g, 3.5 mmol). The reaction mixture was stirred at RT overnight. Then, the reaction mixture was evaporated, and the residue was purified by C18 reverse phase column (Waters XBridge Prep C18 5 µm OBD 19 × 150 mm) on semi-preparative UPLC (Waters 2998 Photodiode Array Detector and 2545 Binary Gradient Module) with a gradient of 3% to 80% methanol in water over 30 min. The overall yields were 50%.

$^1$H-NMR (400 MHz, $d_6$-DMSO): 8.21 (t, 1H), 7.90 (t, 1H), 7.69 (d, 2H), 6.85 (d, 2H), 6.39 (d, 2H), 4.29 (dd, 1H), 4.14 (m, 1H), 3.03 (ddd, 1H), 2.80 (dd, 1H), 2.57 (d, 1H), 2.06 (t, 2H), 1.67–1.19 (m, 6H).

**Fig. 6 | Validation of SCRIB enrichment at the AIS in vitro and in vivo by immunostaining. a** Representative images of DIV14 neurons treated with or without 0.5% Triton X-100 extraction prior to fixation. Fixed hippocampal neurons were stained with SCRIB (red), AnkG (green, AIS), and MAP2 (blue, somatodendrites). Three independent experiments were performed. Scale bar, 50 μm. **b** Representative images of SCRIB knock-out using the CRISPR-Cas9 system. Hippocampal neurons were infected at DIV0 with AAV to express *Scrib* triple gRNA or template plasmids; both also harbor an HA expression cassette. Neurons were fixed at DIV14 and stained for SCRIB (magenta), AnkG (green), HA (red), and MAP2 (blue). Arrows indicate the AIS. Scale bar, 20 μm. **c** Quantification of the percentage of AIS SCRIB positive neurons. Seven independent experiments were performed using

two different *Scrib* triple gRNAs to disrupt *Scrib* expression. The total number of quantified neurons in each condition was shown in the graph. $n = 196$, 184, and 192 neurons analyzed for control, Scrib gRNA1, and Scrib gRNA2, respectively. Data are mean ± SEM. One-way ANOVA, $p = 6.97 \times 10^{-11}$. **d** Comparison of integrated AnkG intensity in the AIS 14 days after transduction with AAV to disrupt *Scrib* expression. Three independent experiments were performed. $n =$ the number of neurons and is shown in the graph. Data are mean ± SEM. Unpaired two-tailed t test followed by Welch's correction; ns, not significant. **e** Coronal cortical sections from P7 mice were stained for SCRIB (green), AnkG (red), and nuclei (DAPI, blue). Three independent experiments were performed. Scale bar, 50 μm.

## Primary neuron culture

Neurons were prepared from cortices or hippocampi of E18.5 embryos or P0 pups. Tissues were dissected and incubated in a digestion solution (0.25% trypsin and 0.4 mg/ml DNase in Ca²⁺/Mg²⁺ free HBSS) at 37 °C for 15 min. Followed by the dissociation, neurons were plated onto 100 μg/ml poly-D-lysine (Sigma) coated 10-cm dishes at a density of 100,000 cells/cm² or on coverslips coated by 20 μg/ml poly-D-lysine and 10 μg/ml laminin mouse protein (Gibco) at a density of 40,000 cells/cm² in plating medium (high glucose DMEM medium containing L-glutamine and 10% fetal bovine serum). After 3 h, the medium was changed into neuronal culture medium (Neurobasal, B27 supplement, Glutamax-I, and penicillin-streptomycin) to maintain neuronal long-term growth. 2 μM cytosine β-D-arabinofuranoside hydrochloride (AraC) was added after 6 days to slow down glia cells proliferation. One-third of culture medium was replaced with fresh neuronal culture medium every seven days.

## Optimization of anti-NFASC antibody directed AIS proximity labeling

Neurons were fixed with 4% formaldehyde for 15 min at RT. The free aldehyde group was quenched by 0.25 M Glycine for 10 min. After washing four times with PBS, endogenous peroxidase of cells was deactivated in 1.5% H₂O₂ as long as 1.5 h. Washed samples were blocked by 5% fetal bovine serum dissolved in PBS containing 0.1% Tween-20 for 1 h at RT. Then neurons were stained with 1, 2, or 5 μg/ml mouse anti-Pan-Neurofascin (anti-NFASC, NeuroMab, Cat# 75-172, stock 1 mg/ml) dissolved in antibody dilution solution (1% FBS with 0.1% Tween-20 in PBS) for 1 h at RT. Followed by washes in washing solutions (0.1% Tween-20 in PBS), samples were labeled with 0.1, 0.2, 0.5, or 1 μg/ml HRP conjugated anti-mouse IgG (Cell signaling technology, Cat# 7076S, stock 1 mg/ml) for 1 h. Samples were washed for another four times with washing solution and incubated with 6 different probes (see Supplementary Fig. 1e) individually at a concentration of 500 μM for 10 min. The proximity labeling was triggered with the addition of H₂O₂ (1, 100, 500, or 1000 μM). After 1, 5, or 10-min reaction, the proximity labeling was stopped by adding quenching solutions (50 mM sodium ascorbate and 5 mM Trolox dissolved in PBS). Endogenous bait protein NFASC and biotinylated proteins were detected by immunocytochemistry to compare proximity labeling specificity and efficiency through microscopy.

## Immunoproximity labeling

The above optimized parameters were applied to anti-NFASC, anti-NeuN, anti-MAP2, and anti-SMI312 directed proximity labeling to capture AIS, soma, somatodendrites, and axon components. The blocked samples were stained with primary antibodies dissolved in antibody dilution solution (1% FBS with 0.1%Tween-20 in PBS) for 1 h at RT. After washing, samples were labeled with HRP conjugated second antibodies for 1 h and then incubated with 500 μM biotin-phenol (BP) for 10 min. The proximity labeling was triggered with the addition of 100 μM H₂O₂. After 1 min reaction, free radicals were quickly quenched twice by quenching solutions (50 mM sodium ascorbate and 5 mM Trolox dissolved in PBS). Samples were finally washed with PBS.

In the case of anti-AnkG and anti-TRIM46 targeted proximity labeling, parameters including antibody dilutions and H₂O₂ concentration were also individually tested. Antibodies used for proximity labeling were listed as follows. Primary antibodies: mouse anti-Pan-Neurofascin (anti-NFASC, NeuroMab, Cat# 75-172), mouse anti-AnkG (NeuroMab, Cat# 75-146), mouse anti-NeuN (Abcam, Cat# ab104224), mouse anti-MAP2 (Sigma-Aldrich, Cat# M1406), mouse anti-Neurofilament (anti-SMI312, Biolegend, Cat# 837904), chicken anti-AnkG (Synaptic Systems, Cat# 386006) and chicken anti-TRIM46 (Synaptic Systems, Cat# 377006). HRP conjugated second antibodies: anti-mouse IgG (Cell signaling technology, Cat# 7076S) or anti-chicken IgY (Invitrogen, Cat# A16054; Abcam, Cat# ab6877).

## Protein lysis of cortical neurons after proximity labeling

Dimethyl experiments include three pairs: anti-NFASC vs. no primary antibody; anti-NFASC vs. anti-NeuN; and anti-NFASC vs. anti-MAP2. Three times independent replicates were performed in each pair. The 10-plex TMT DIV14 MS experiment include: three biological replicates for anti-NFASC targeted proximity labeling; no primary antibody or BP probe negative controls; reference samples of anti-NeuN, anti-MAP2 (duplicates), and anti-SMI312 (duplicates). Due to differences in the bait abundance, reference samples of anti-NeuN, anti-MAP2 and anti-SMI312 yielded higher amounts of labeled proteins than the anti-NFASC sample. To balance peptide loading to LC-MS/MS analysis, we used only a fraction of the reference samples for biotinylated proteins enrichment. Anti-NF186, no primary antibody and no substrate BP shared the same protein quantity for enrichment. 1/2 protein quantity was used for anti-NeuN enrichment and 1/4 amount was used for anti-MAP2 and anti-SMI312 enrichment, respectively.

In AIS developmental TMT experiments, two parallel 10-plex TMT experiments were performed and DIV14 samples were used as a reference for DIV7 and DIV21 samples. Proximity labeled samples at DIV14 were equally divided and used for TMT1 and TMT2 pipeline. Five conditions were designed for each stage: anti-NFASC targeted AIS proximity labeling (duplicates); no primary antibody (-1Ab); anti-NeuN targeted soma proximity labeling and anti-MAP2 targeted somatodendrites proximity labeling. An amount of 1.5 mg proteins in samples of anti-NFASC and no primary antibody (DIV7, 14 and 21) were used for enrichment, while 1/2 and 1/4 protein quantity were used for anti-NeuN and anti-MAP2 enrichment individually.

Proximity labeling samples were lysed with lysis solution (50 mM Tris-HCl pH 7.6, 150 mM NaCl, 1% SDS, 0.5% sodium deoxycholate, 1% Triton X-100 and protease inhibitors cocktail) on ice for 10 min. 500 μl lysis solution was added for each 10-cm dish. In the condition of anti-NFASC, no primary antibody and no substrate BP, three 10-cm dishes in one replicate were needed for protein lysis. While for anti-NeuN, anti-MAP2, and anti-SMI312 conditions, one or two 10-cm dishes were needed. Protein lysis was de-crosslinked at 99 °C for 1 h. The lysed solution was sonicated on ice and then centrifuged at 12,000 × g for 10 min at 4 °C to collect supernatant. Proteins were precipitated in cold methanol at −80 °C.

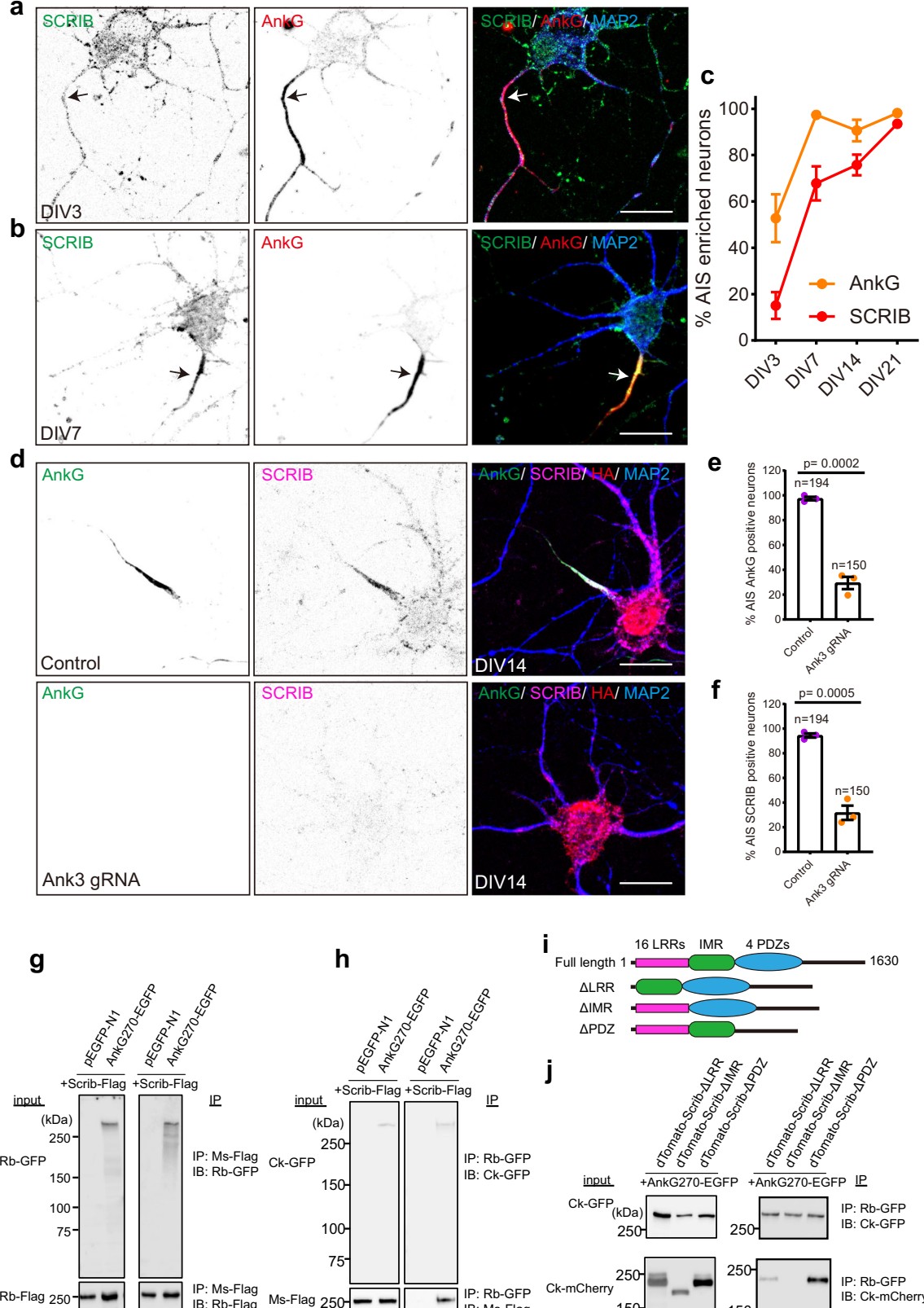

## Enrichment of biotinylated proteins and on-bead digestion

Precipitated proteins were washed twice using cold methanol. Purified proteins were dissolved completely in 0.5% SDS (w/v). Protein concentrations were measured using a BCA Protein Assay Kit to adjust protein quantity for enrichment. 1% of pre-enrichment proteins were kept to test biotinylated signal via Western blotting or silver staining to assess proximity labeling efficiency. The remaining proteins were incubated with 200 µl streptavidin beads for 3 h with gentle rotation at RT. 5% of post-enrichment proteins were kept to test the enrichment efficiency via Western blotting or silver staining. Then the protein-beads mixture was sequentially washed twice by 2% SDS (w/v), 8 M urea, and 2 M sodium chloride. The mixture was reduced by 10 mM

**Fig. 7 | AnkG is required for SCRIB enrichment at the AIS. a, b** Representative images of DIV3 (**a**) and DIV7 (**b**) hippocampal neurons stained for SCRIB (green), AnkG (red), and MAP2 (blue). Arrows indicate the proximal axon (**a**) and AIS (**b**). Scale bar, 20 μm. **c** Quantification of the percentage of neurons with AIS SCRIB and AnkG at DIV3, 7, 14, and 21. Four independent experiments were performed at each time point. Data are mean ± SEM. **d** Representative images of AnkG knock-out using the CRISPR-Cas9 system. DIV0 hippocampal neurons were transduced with *Ank3* triple gRNA or template plasmids; the plasmids also harbor an HA expression cassette. Neurons were fixed at DIV14 and stained for SCRIB (magenta), AnkG (green), HA (red), and MAP2 (blue). Scale bar, 20 μm. **e, f** Quantification of the percentage of neurons with AIS AnkG (**e**) and SCRIB (**f**) after loss transduction with AAV to disrupt AnkG expression. Three independent experiments were performed

with the number of neurons analyzed indicated on the figure. Data are mean ± SEM. Unpaired two-tailed t test, *p* = 0.0002 (**e**) and *p* = 0.0005 (**f**). **g, h** Co-immunoprecipitation of Scrib-Flag with AnkG270-EGFP. Scrib-Flag and AnkG270-EGFP or pEGFPN1 were co-transfected in HEK293T cells and immunoprecipitated by Flag antibody (**g**) or GFP antibody (**h**). Two independent experiments were performed. IP immunoprecipitation, IB immunoblotting, Rb-GFP rabbit anti-GFP, Rb-Flag rabbit anti-Flag, Ms-Flag mouse anti-Flag, Ck-GFP chicken anti-GFP. **i** Illustration of the full length and truncated SCRIB constructs. **j** Co-immunoprecipitation of truncated SCRIB with AnkG270. The experiment was performed once. IP immunoprecipitation, IB immunoblotting, Ck-GFP chicken anti-GFP, Ck-mCherry chicken anti-mCherry, Rb-GFP rabbit anti-GFP.

dithiothreitol and then with 20 mM iodoacetamide for alkylation. After washing four times with 100 mM TEAB, proteins on beads were digested with the addition of 2 μg trypsin (Promega, Cat# V511A) for 18 h at 37 °C.

## Immunoblotting and silver staining

Protein samples were heated at 99 °C for 10 min with the addition of reducing Laemmli SDS loading buffer. In the case of post-enrichment biotinylated protein samples, 2 mM biotin was added to elute biotinylated proteins. In the case of co-immunoprecipitation (IP) experiments, AnkG270-EGFP or pEGFPN1 were co-transfected with Scrib-Flag or its truncated constructs in HEK293T cells (a gift from Dr Benjamin Arenkiel in BCM) and lysed after 48 h. The Scrib-Flag plasmid was generated by amplifying human *Scrib* coding sequence from MSCV Puro SCRIB WT (Addgene, Cat# 88886) and fused with 3xFlag tag at the C-terminal in pcDNA3 backbone. AnkG270-GFP was a kind gift from Dr. Vann Bennett (Duke University). The *Scrib* truncation mutants dTomato-Scrib-ΔLRR, dTomato-Scrib-ΔIMR, and dTomato-Scrib-ΔPDZ were kind gifts from Maya Shelly (Stony Brook University).

Input and immunoprecipitated samples were dissolved in Laemmli buffer (62.5 mM Tris-HCl (pH 6.8), 2% SDS, 10% Glycerol, 2% 2-Mercaptoethanol, and 0.005% Bromophenol blue) for immunoblotting. Proteins were loaded for electrophoresis in 5% SDS-PAGE stacking gel. 10% or 6% SDS-PAGE separating gels were used for biotinylated samples and co-IP samples, respectively. The blots were blocked with 5% BSA at RT for 1 h and then sequentially labeled using primary antibodies and HRP conjugated secondary antibodies. Antibodies used were: rabbit anti-GFP (Invitrogen, Cat# A11122, 1:1000), chicken anti-GFP (Aves Labs, Cat# GFP-1020, 1:2000), mouse anti-Flag (MBL, Cat# M185-3L, 1:1000), chicken anti-mCherry (Aves Labs, Cat# MCHERRY-0020, 1:1000), rabbit anti-Flag conjugated with HRP (Cell Signaling, Cat# 86861, 1:1000), streptavidin-HRP (Invitrogen, Cat# 21124, 1:5000), goat anti-mouse IgG (H + L) conjugated with HRP (Jackson, Cat# 115-035-146, 1:5000), goat anti-rabbit IgG (H + L) conjugated with HRP (Jackson, Cat# 111-035-003, 1:5000), and goat anti-chicken IgY conjugated with HRP (Aves, Cat# H-1004, 1:5000). The blots were developed by Clarity western ECL substrate (Bio-Rad, Cat# 1705060) and imaged in ChemiDoc MP Imaging System (Bio-Rad). In addition, the SDS-PAGE gels could be directly used for silver staining using the Fast Silver Stain Kit (Beyotime, Cat# P0017S).

## Dimethyl labeling

Enriched proteins were digested with trypsin and the resulting peptides were treated with isotope-encoded formaldehyde (heavy $D^{13}CDO$ for anti-NFASC samples and light HCHO for negative control/reference samples) and NaBH₃CN for their −NH₂ groups methylation, resulting in a mass shift from 34.0631 Da to 28.0313 Da. For pairs of anti-NFASC and no primary antibody, peptides were combined directly and desalted by C18 tips (ThermoFisher, Cat#87784). For pairs of anti-NFASC/anti-NeuN or anti-NFASC/anti-MAP2, peptides were desalted first separately and then equal amounts of peptides were combined. Peptides concentration was measured by Quantitative Colorimetric

Peptide Assay Kit (ThermoFisher, Cat# 23275). Combined peptides were dried in a vacuum concentrator and ready for LC-MS/MS.

## 10-plex TMT labeling and peptides fractionation

Digested peptides were desalted by C18 tips (ThermoFisher, Cat#87784). In the case of 14 DIV independent TMT experiments, desalted total peptides of anti-NF186, no primary antibody and no substrate BP were directly used for TMT labeling. Taking anti-NF186 peptides amount as reference, the same quantity of peptides from anti-NeuN, anti-MAP2, and anti-SMI312 conditions were prepared in advance. Peptide concentration was measured by Quantitative Colorimetric Peptide Assay Kit (ThermoFisher, Cat# 23275). In the case of AIS developmental TMT experiments, sample amount was controlled when performing enrichment. Therefore, these peptides were directly applied to TMT labeling.

TMT labeling was performed as instructed by TMT 10-plex Mass Tag Labeling Kits and Reagents (ThermoFisher, Cat# 90110). Briefly, TMT reagents were maintained at RT and reconstituted with 41 μl anhydrous acetonitrile for 0.8 mg vial of each tag. Peptides were reconstituted in 10 μl 100 mM TEAB and dissolved completely. Add 12.5 μl TMT solution into dissolved peptides and label for 2 h in the dark. The labeling in each condition was described in Figs. 2b and 3a. The reaction was quenched by adding 8 μl 5% hydroxylamine and which was incubated for 15 min. Peptides from ten conditions were mixed and dried in a vacuum concentrator. Mixed peptides were re-dissolved in 300 μl 0.1% trifluoroacetic acid and fractionated according to the instructions of Pierce High pH Reversed-Phase Peptide Fractionation Kit (ThermoFisher, Cat# 84868). Samples were eluted by gradient acetonitrile of 10.0%, 12.5%, 15.0%, 17.5%, 20.0%, 22.5%, 25.0%, 50.0% solution in 0.1% triethylamine and defined as fraction 1−8, respectively. Fractions were combined as pairs of '1 + 5', '2 + 6', '3 + 7', and '4 + 8' and dried again in vacuum concentrators. Peptides were ready for LC-MS/MS.

## Liquid chromatography and mass spectrometry

Peptides were reconstituted in 0.1% formic acid and separated in a loading column (100 μm × 2 cm) and a C18 separating capillary column (75 μm × 15 cm) packed in-house with Luna 3 μm C18(2) bulk packing material (Phenomenex, USA). A 2-h liquid chromatography (LC) was applied to peptides separation. The LC gradient was held at 2% for the first 8 minutes of the analysis, followed by an increase from 2% to 10% B from 8 to 9 minutes, an increase from 10% to 44% B from 9 to 123 minutes, and an increase from 44% to 99% B from 123 to 128 min (A: 0.1% formic acid in water and B: 80% acetonitrile with 0.1% formic acid).

Samples were analyzed by Orbitrap Fusion LUMOS Tribrid Mass Spectrometer. The precursors were ionized using an EASY-Spray ionization source (Thermo Fisher Scientific). Survey scans of peptide precursors were collected in the Orbitrap from 350–1600 Th with an advance gain control (AGC) target of 400,000, a maximum injection time of 50 ms, RF lens at 30%, and a resolution of 60,000 at 200 *m/z*. Monoisotopic precursor selection was enabled for peptide isotopic distributions, precursors of z = 2−7 were selected for data-dependent

MS/MS scans for 3 s of cycle time, and dynamic exclusion was set to 15 s with a ±10 ppm window set around the precursor mono-isotope. In HCD scans, an automated scan range determination was enabled. An isolation window of 1.6 Th was used to select precursor ions with the quadrupole. Product ions were collected in the Orbitrap with the first mass of 110 Th, an AGC target of 50,000, a maximum injection time of 30 ms, HCD collision energy at 30%, and a resolution of 15,000.

## Mass spectrometry data analysis

Raw data files were loaded in MaxQuant software (version 1.6.10.43) and searched against *Rattus norvegicus* proteomes from Uniprot database (Taxonomy, 10116, downloaded on Nov 15th, 2020). Trypsin was selected as the digestion mode. In the case of searching dimethyl proteomics, the composition of $Hx(6)C(2)H(-2)$ was replaced by $Hx(4)Cx(2)$ in heavy labels of DimethLys6 and DimethNter6. Cysteine acetylation (carbamidomethyl) was a fixed modification. Methionine oxidation and protein N-terminal acetylation were variable modifications. Re-quantify button in Misc. module and match between runs in Identification module were activated. In the case of 10-plex TMT proteomics, reporter ion MS2 was selected and 10-plex TMT was activated. Correction factors were added to the tags based on the reagent instructions (ThermoFisher, Cat# 90110, Lot: UL291038). In both cases, the false discovery rate was set to 0.01.

## Proteome data analysis

Anti-NFASC targeted AIS proximity labeling was the positive group. Samples of no primary antibody or no substrate BP were negative controls for cutoff analysis, while anti-NeuN, anti-MAP2, or anti-SMI312 targeted soma, somatodendrites, or axon proximity labeling were reference controls for ranking AIS candidates. Top-ranked proteins would have more possibility to be expressed or enriched in the AIS.

In the case of dimethyl experiments, contaminants and proteins identified as reversed hits were removed. Proteins containing at least 2 unique peptides and quantified in at least two out of three independent replicates were retained for further analysis (anti-NFASC vs. anti-NeuN: 965; anti-NFASC vs. anti-MAP2: 802). In addition, an H/L ratio >1.5 was set as the cutoff for each replicate in the experiment of anti-NFASC versus no primary antibody, which resulted in the identification of 704 proteins. Proteins in three pairs were intersected and a total of 568 proteins were obtained. We normalized the H/L ratio against the medium in each column using the formula of '$\log_2(H/L)$-medium $\log_2(H/L)$' in +NFASC/+NeuN and +NFASC/+MAP2 replicates. Averaged normalized data was obtained from three replicates and used for distribution analysis.

In the case of DIV14 TMT experiments, contaminants and proteins identified as reversed hits were removed. Proteins with at least two unique peptides were kept. Ratios were calculated by division using their MS intensity. Parallel analysis was analyzed using Pearson correlation. The average ratio of replicates was used for the following analysis. The cutoff ratio was set at 2 for +NFASC/-BP and +NFASC/-1Ab. A total of 1403 proteins were retained for further analysis. Ratios of +NFASC/+NeuN, +NFASC/+MAP2, and +NFASC/+SMI312 were normalized against the medium ratio in their corresponding column. Proteins were ranked based on average normalized $\log_2$(+NFASC/+References, fold change) in a descending order. The final overall rank was based on the average rank scores against soma, somatodendrites, and axon. Proteins in the top 200 against each reference were overlapped and a total of 71 proteins were obtained. Gene Ontology analysis of these 71 proteins was performed PANTHER overrepresentation test. A total of 2755 neuronal proteins in this TMT were used as a reference. Among them, 2744 proteins were valid data in Gene Ontology database. Top 10 cellular compartment terms were presented.

In the case of AIS developmental TMT experiments, two parallel 10-plex TMT labeling were employed to analyze AIS developing (DIV7), immature (DIV14) and completely mature (DIV21) proteome changes. Contaminants and proteins identified as reversed hits were removed. Proteins with at least two unique peptides were kept. Ratios were calculated by division using their MS intensity. Parallel analysis was analyzed using Pearson correlation. The average ratio of replicates was used for further analysis. Proteins were filtered with the standard of 'average $\log_2$(+NFASC)-$\log_2$(-NFASC) > 3.5' for DIV7 and 21, while for DIV14 the standard was set as 'average $\log_2$(+NFASC)-$\log_2$(-NFASC) > 1'. This setting was based on the endogenous biotinylated proteins distribution. Proteins with intensity in +NFASC condition, not in -NFASC condition were also retained. Ratios of +NFASC/+NeuN and +NFASC/+MAP2 were normalized against the medium ratio in their corresponding column in each time point. Proteins were ranked based on average normalized $\log_2$(+ NFASC/+References, fold change) in a descending order. The final overall rank was based on the average rank scores against soma and somatodendrites.

For analysis of AIS protein dynamics along neuronal development, biotinylated proteins at DIV7, 14 and 21 together with prior DIV14 biotinylated proteins were overlapped and revealed 549 common proteins in three stages. These proteins in the condition of anti-NFASC at DIV7, 14 and 21 were normalized based on endogenous biotinylated protein PCCA[54]. Among 549 proteins, 534 proteins were presented more than 20% changes of average MS intensity in at least one pair of time points. Heatmap was performed using cutree script in R package of pheatmap. AIS molecules identified in three stages were performed Z-scored normalized abundance analysis. Significant analysis and fold changes were analyzed regarding these AIS proteins according to a guide of statistics for proteomics data analysis[55].

## Immunocytochemistry and immunohistochemistry

For immunocytochemistry, neurons were fixed in 4% formaldehyde for 15 min at RT and washed three times in PBS. Samples were blocked by 5% fetal bovine serum dissolved in PBS containing 0.1% Tween-20 for 1 h at RT. Then neurons were sequentially stained with primary antibodies and fluorescent second antibodies for 1 h at RT. Nuclei were stained with DAPI (Invitrogen, Cat# D1306). In the case of detergent extraction experiments, neurons were treated with 0.5% Triton X-100 in PBS for 5 min at RT before fixation.

Primary antibodies used for immunocytochemistry in this study were: mouse anti-Pan-Neurofascin (NeuroMab, Cat# 75-172, 1:1000), mouse anti-AnkG (NeuroMab, Cat# 75-146, 1:1000), chicken anti-AnkG (Synaptic Systems, Cat# 386006, 1:500), guinea pig anti-AnkG (Synaptic Systems, Cat# 386004, 1:1000), mouse anti-NeuN (Abcam, Cat# ab104224, 1:1000), mouse anti-Neurofilament (Biolegend, Cat# 837904, 1:1000), mouse anti-MAP2 (Sigma-Aldrich, Cat# M1406, 1:1000), chicken anti-MAP2 (Abcam, Cat# ab5392, 1:10,000), chicken anti-MAP2 (EnCor Biotechnology, Cat# CPCA-MAP2, 1: 2000), chicken anti-TRIM46 (Synaptic Systems, Cat# 377006, 1:500), rabbit anti-βIV-spectrin (laboratory of Dr. Matthew Rasband in BCM, 1:1000), rabbit anti-SCRIB (Abclonal, Cat# A17450, 1:50), rabbit anti-SCRIB (Invitrogen, Cat# PA5-28628, 1:100), mouse anti-V5 (Invitrogen, Cat# R960CUS, 1:500) and rat anti-HA (Millipore Sigma, Cat# 11867423001, 1:1000). Fluorescent second antibodies were: goat anti-mouse IgG2a, Alexa Fluor 488 (Invitrogen, Cat# A21131, 1:1000); goat anti-mouse IgG (H + L), Alexa Flour 488 (Invitrogen, Cat# A11029, 1:1000), 568 (Invitrogen, Cat# A11031, 1:1000), and Plus 594 (Invitrogen, Cat# A32742, 1:1000); goat anti-rabbit IgG (H + L), Alexa Flour 488 (Invitrogen, Cat# A11034, 1:1000), and 568 (Invitrogen, Cat# A11036, 1:1000); goat anti-rat IgG (H + L), Alexa Flour 594 (Invitrogen, Cat# A11007, 1:1000); goat anti-chicken IgY (H + L), Alexa Flour 488 (Invitrogen, Cat# A11039, 1:1000), 647 (Invitrogen, Cat# A21449, 1:1000), and AMCA (Jackson, Cat# 103-155-155, 1:200). In the case of proximity labeling samples, biotinylated protein signal was detected by incubation with streptavidin conjugated with Alexa Fluor 568 (Invitrogen, Cat# S11226, 1:1000) or 647 (Invitrogen, Cat# S21374, 1:1000).

For immunohistochemistry, brain sections of P7 C57BL/6 mice or P23 Cas9 mice tagged with smFP-V5 to *Scrib* were used for the staining. Cortical sections were blocked for 2 h at RT in 10% fetal bovine serum dissolved in PBS containing 0.1% Triton X-100 and then incubated with first antibodies at 4 °C overnight. After washing, slices were incubated with second antibodies and counterstained with DAPI. First antibodies used were: mouse anti-AnkG (NeuroMab, Cat# 75-146, 1:500), rabbit anti-SCRIB (Abclonal, Cat# A17450, 1:50), mouse anti-V5 (Invitrogen, Cat# R960CUS, 1:500), and rabbit anti-βIV-spectrin (laboratory of Dr. Matthew Rasband, 1:500). Second antibodies used were: goat anti-mouse IgG (H + L), Alexa Flour 568 (Invitrogen, Cat# A11031, 1:1000), goat anti-rabbit IgG (H + L), Alexa Flour 488 (Invitrogen, Cat# A11034, 1:1000), and 568 (Invitrogen, Cat# A11036, 1:1000), and goat anti-mouse IgG2a, Alexa Fluor 488 (Invitrogen, Cat# A21131, 1:1000). Sections were mounted to glass slides with Fluoromount-G (SouthernBiotech, Cat# 0100-01).

Images were acquired on an inverted fluorescence microscope (Nikon-TiE, Japan) equipped with a 40×1.3 NA oil immersion objective lens, confocal laser lines (OBIS 405, 488, 532, 561, and 637 nm, Coherent, USA), spinning disk confocal unit (Yokogawa CSU-X1, Japan) and scientific CMOS cameras (Hamamatsu ORCA-Flash 4.0 v2). The microscope, camera, and lasers were controlled with a customer-built software written in LabVIEW (National Instruments). Sections were imaged in the focal position or in z-stacks with a 0.5 μm step size at a resolution of 1500 × 1000 pixels. Z-projection was obtained in Fiji-ImageJ software. Figures were adjusted and prepared in Fiji-ImageJ and Adobe Photoshop CC2018.

### V5 tag knock-in and microscopy

V5 tag knock-in sgRNA and donor constructs were generated following previously described strategies[44,56]. The specific target sequences of guide RNA (gRNA) were designed for knock-in: rat *Scrib* 5′-GGGAAG-CACCTGGCCCTAGG-3′; rat *Wdr47* 5′-TCTGGACTTACAGTGGCTAG-3′; rat *Wdr7* 5′-ACGGCAGTCAGACCATGAAG-3′; mouse *Scrib* 5′-GTCCCGGCCGCACAGACCGA-3′. Plasmids of gRNA or AAV-SpCas9 (Addgene# 60957, Dr. Feng Zhang) together with pUCmini-iCAP-PHP.S (Addgene# 103002, Dr. Viviana Gradinaru) and pHelper (Cat #240071, Agilent Technologies) were used for AAV production. Adeno-associated virus (AAV) cell-lysates were produced using the AAVpro Purification Kit (All Serotypes) (Takara) with slight modifications. Briefly, HEK293T cells were transfected with AAV plasmid, helper plasmid, and serotype PHP.S plasmid and collected after three days. The cells were lysed using the AAV Extraction Solution A plus and B, and this crude AAV solution was used for neuronal in vitro transduction. An aliquot of 20 μl AAVs (10 μl of Cas9 and 10 μl of knock-in sgRNA and donor) was added to a well of 12-well plates at DIV0. The medium was replaced after two days of infection. Neurons were fixed at the indicated time points for immunocytochemistry. Concentrated AAV solution of smFP-V5 knock-in to mouse *Scrib* was used for AAV intra-ventricular injection on P0 Cas9 mice pups. 50–100 μl AAV was concentrated from 2 × 15-cm dishes and 2 μl was used for each hemisphere.

Imaging was performed using a Zeiss AxioImager Z2 fitted with apotome for structured illumination. Images were captured using the Zeiss ZEN software. Super-resolution images were performed by Stimulated Emission Depletion (STED) microscope using a STEDyCON (Abberior) system fitted to a Nikon Eclipse Ti2 microscope. The periodicity of SCRIB was measured in Fiji-ImageJ after generating a line scan across the AIS. The distance between peak fluorescence intensities (maximum to maximum) was then measured.

### Genes knock-out and imaging by microscopy

Triple gRNAs for rat *Scrib* and *Ank3* were designed: *Scrib* gRNA1 (5′-GCTATTGAACTTGCGGAAGT-3′, 5′-CCAGGCATACCAGCCGCCGC-3′, and 5′-CTGTGAGGATCAGTTCCGAG-3′); *Scrib* gRNA2 (5′-AGCC ACAGCTCCCGTAGGTT-3′, 5′-GGCTAACTTCATGCAACTGG-3′, and

5′-TTATGCTCTCAGGTATCTCG-3′); *Ank3* gRNA (5′-CTGCTCGAGAA CGACACGAA-3′, 5′-CGCTCGGTTTAACAGCAACG-3′, and 5′-CTTCAC GCCGCTGTATATGG-3′). The plasmid backbone consists of a cassette of synapsin1 promoted tandem HA tag as indicator for transfection. Crude AAV virus were generated by co-transfection of AAV plasmid, helper plasmid, and serotype PHP.S plasmid in HEK293T cells. An aliquot of 20 μl AAVs (10 μl of Cas9 and 10 μl of knock-out sgRNAs) was added to a well of 12-well plates at DIV0. Neurons were fixed at DIV14 and stained for imaging using a Zeiss structured illumination microscope with apotome.

### Image analysis

Cortical neurons of DIV10-14 were processed for optimizing AIS proximity labeling parameters. AIS was defined by endogenous NF186. Biotinylated proteins signal in the AIS and soma was quantified. Images were taken using the same parameter settings in all the conditions of every experiment. Correct mean intensity (CMF) was obtained using mean intensity in target regions extracted by the background of the same image. Labeling efficiency was evaluated by the AIS CMF. Labeling specificity was evaluated by the ratio of AIS CMF/Soma CMF.

smFP-V5 tag knock-in to SCRIB, WDR47 and WDR7 samples at DIV14 were used for polarity analysis. Neurons were infected at DIV0. The AIS was defined by βIV-spectrin labeling. V5 signal was measured in the AIS and proximal dendrites (similar width). A 10-pixel line was drawn along the AIS, or the proximal dendrites to measure the intensity using Fiji-ImageJ software (NIH). Background was extracted and the polarity index was evaluated by the formula AIS/Dendrite ratio = AIS CMF/Dendrite CMF. At least three independent experiments were performed.

DIV14 neurons were used for the integrated AnkG intensity measurement at the AIS. Images were acquired in the focal position for quantification. A 10-pixel line was drawn along the AIS using Fiji-ImageJ software. Integrated AnkG intensity was obtained and normalized against control. Three times independent experiments were performed.

### Statistical analysis

All statistical analyses were performed in GraphPad Prism 8. The statistical details including the number of experiments, number of cells, and statistical tests can be found in figure legends. Statistical analysis was performed by two-tailed t-test for two group comparisons and by one-way ANOVA for multiple group comparisons. Graphs are presented as the mean ± SEM. Differences were considered significant when *p*-values were less than 0.05.

### Reporting summary

Further information on research design is available in the Nature Portfolio Reporting Summary linked to this article.

## Data availability

All data generated or analyzed during this study are included in this published article. Raw data of proteome files have been deposited in the ProteomeXchange database under the accession code PXD045921. Original proteomics datasets from MaxQuant and analyzed source data underlying related figures in this study are included as Supplementary Data 1–3 and a source data file. Source data are provided with this paper.

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

## Acknowledgements

This work was supported by the Ministry of Science and Technology 2022YFA1304700 (P.Z.), 2018YFA0507600 (P.Z.), 2017YFA0503600 (P.Z.), the National Natural Science Foundation of China 32088101 (P.Z.), 21727806 (P.Z.), a grant from the National Institutes of Neurological Disorders and Stroke NS122073 (M.N.R.), and the Dr. Miriam and Sheldon G. Adelson Medical Research Foundation (M.N.R.). P.Z. is sponsored by Bayer Investigator Award. W.Z. was supported in part by the Postdoctoral Fellowship of Peking-Tsinghua Center for Life Sciences. We thank Yi Li, Gang Wang, Feng Yuan, Xin Zeng, and Yu Chen for helpful discussions. We thank Ms. Wen Zhou from the Analytical Instrumentation Center in Peking University for assistance with MS sample identification.

## Author contributions

W.Z., P.Z., and M.N.R. conceived the project. W.Z., P.Z., and M.N.R. designed experiments. W.Z., Y.F., L.P., Y.O., X.D., A.R., X.Z., and M.S. performed experiments. W.Z. and P.Z. analyzed the data and wrote the paper with input from all authors. W.Z., P.Z., and M.N.R. revised the paper.

## Competing interests

The authors declare no competing interests.
