## [Peer Review File · Nature Communications]

REVIEWER COMMENTS

Reviewer #1 (Remarks to the Author):

This paper uses a novel proteomic approach to characterise the AIS proteome in cultured neurons through development, and to identify some potentially important new AIS-localised components. This is a valuable dataset which appears extremely carefully produced – the comparison of AIS-localised vs soma/dendrite/axon-localised proteins is especially powerful. The authors also present data from pharmacological manipulations suggesting that PHGDH activity is important for the structural and functional integrity of the mature AIS. I have a few relatively minor concerns regarding these latter experiments, but overall the standard of data collection and presentation is high and the results largely convincing.

1) Electrophysiological recordings. Numerous quality control measures and methodological details for the patch-clamp recordings are missing from the manuscript. The authors should provide details of mean±-SEM series resistance and passive membrane properties (R_{in}/C_m) for each group – did these differ? What was the maximum permitted R_s , and permitted change in R_s during a recording? Was fast/slow capacitance compensation applied? In the current-clamp recordings, was bridge balance used? The example traces in Fig 6B NCT suggest not, and this may have introduced unfortunate variability in voltage measures such as AP threshold & amplitude – indeed, this could be a reason why no significant difference in AP threshold was observed when this is usually the clearest functional correlate accompanying a change in AIS length. The effect of NCT on max rising speed is probably the most convincing (and certainly the effect here that is most likely to be linked to AIS changes), but the authors should show some example AP phase-plane plots to illustrate this difference more clearly, and should provide details in the Methods as to how exactly this measure was calculated. It also appears that all current clamp recordings were obtained from the resting holding voltage (i.e. $I=0$) rather than from a constant baseline – this is problematic because of the effect of NCT on RMP, which almost certainly accounts for the increase in rheobase. Does RMP correlate with rheobase density (or any other measure, especially max rising phase) on a cell-by-cell level? And why is rheobase expressed as a density (probably appropriately) but input current in Fig6H is not? Ideally, all current clamp recordings should be re-done with 100% bridge balance and from a constant holding voltage. If this is not feasible, the authors should be explicit in the manuscript about how these factors may have influenced their data, and/or should provide additional analyses that show they are not influential (e.g. a lack of correlation between RMP and rheobase/max rising phase). The voltage-clamp I_{Na} experiments are unfortunately entirely unconvincing, and I strongly suggest they are removed from the manuscript. In the absence of any example traces, I find it highly unlikely that fast voltage-gated sodium currents were effectively clamped under standard recording conditions with high external $[Na]$ and no apparent series resistance compensation. Finally, the authors need to modify their claim in the Abstract (line 27) that ‘neuronal PHGDH activity is necessary for AIS integrity and action potential initiation’, given that APs can actually be readily initiated after chronic PHGDH inhibition.

2) The example MAP2 images are not convincing evidence that cell health is unaffected by PHGDH block. The hyperpolarised RMP in the NCT group (Fig 6C) is actually much more convincing (unhealthy cells tend to have depolarised RMP). A lack of difference in passive membrane properties (see above) would also add weight here.

3) AIS start and end positions were estimated by eye, and there is no mention of experimenters being blind to treatment group. Can the authors provide reassurance that their AIS measurements are reliable and unbiased? For example, do they have strong intra- and inter-experimenter reproducibility? Also, with such subjective measures, might the reported AIS length changes be a direct consequence of decreased labelling intensity? Finally, why were AIS length values normalised? I completely see the value in normalising intensity measures (though please describe in the Methods exactly how this normalisation was carried out), but it would be much clearer if absolute AIS length values were reported and tested. If AIS lengths needed to be normalised because of culture-to-culture (or coverslip-to-coverslip) variation in labelling intensity, this suggests that a means of measuring AIS length needs to be found that is independent of label strength.

4) The AIS localisation measures in Fig 4CDIJ are very welcome, but it would be great if these could also be applied to the P2 labelling (Fig S5) and the KI labelling (Fig4F) to allow quantitative comparisons between developmental stages and different labelling approaches. Also, please provide details in the Methods of whether these are measure of maximum or integrated intensity (and if the latter, over which range?), and how exactly background fluorescence levels were determined.

5) A 2-way statistical test is required for the data in Fig 5M-P (effect of drug; effect of astrocytes; effect of drug-x-astrocyte interaction)

6) Line 100: the authors performed a detailed optimisation of all the parameters in the proximity labelling system. Both the optimal numbers for H₂O₂ concentration (100 μ M) and the time (1 minute) are mentioned in the text, but not the chosen concentrations for primary and secondary antibodies. What does 'proper dilution of antibodies' mean? What was prioritised for these? Was it a combination between an acceptable ratio of biotin detected and specificity? Or only specificity?

7) Figure legend Figure S1: please, specify how many cells were analysed for the graphs. Is each dot one cell?

8) Line 137 and Figure 1 legend: add the stage of the neurons ('DIV14') to the figure legend in D to keep the same style within the legend annotation.

9) Figure 1C: please, add an indicator for molecular weight in the blots.

10) Figure S3: please, add an indicator for molecular weight in the blots.

11) The authors claim that the newly found AIS proteins PHGDH colocalizes with AnkG in cultured neurons (line 264: 'TritonX-100 treatment revealed a detergent-resistant pool of PHGFH that colocalized with AnkG at the AIS (Figure 4E)). A series of AIS fluorescence linescans proving such colocalization would help backing up that claim.

12) Figure 4E and 4K: please, indicate where the cell body is in these images.

13) Figure legend of figure S1: in line with the rest of figure legends, please include the chosen statistical test at the bottom of the legend.

14) In Methods, the section 'Detergent extraction' (line794) could go to 'Immunocytochemistry and immunohistochemistry' instead (line 746).

15) Methods, line 607: please, provide the details about the protein loading buffer composition.

16) More details about how the periodicity was measured in figure 4 are needed in the methods section.

Reviewer #2 (Remarks to the Author):

The authors have established the APL-AIS, a novel biotinylated labeling method for comprehensive profiling of molecular architectures around AIS more extensively and sensitively than conventional methods. By applying it mainly to cultured neurons, the authors have proposed a novel molecular mechanism of AIS formation and function that is presumably mediated by PHGDH, whose involvement is previously totally unknown.

AIS-mediated action potential generation is one of the most fundamental events in the neural network of the brain, and elucidation of its molecular architecture and function will provide fundamental knowledge for understanding neural plasticity and neural circuit dysfunction in neurological diseases.

Genetic mutations in enzymes involved in de novo L-serine synthesis in the brain induce severe neurodevelopmental defects, and in recent years, abnormalities in the L-serine synthesis pathway and its involvement in the pathogenesis of various neurodegenerative diseases including Alzheimer's disease have been postulated. From this perspective, this manuscript, which proposes a contribution of L-serine synthase PHGDH in AIS formation, is a study that may bring conceptual novelty to the link between neurotransmission and metabolites.

However, there are concerns regarding methodology and data interpretation that cannot be ignored and should be resolved. Specifically, concerns need to be resolved regarding the two main novel findings the authors claim, the localization of PHGDH to the neuronal AIS and the contribution of serine synthesis by PHGDH in the assembly and function of the AIS.

I. Localization of PHGDH to the AIS in neurons

First, there are problems that the authors do not mention regarding the experimental conditions of cultured cells used in this study. Primary neuronal cultures used for this study were prepared from cerebral cortex or hippocampus on the day of birth (postnatal day 0) and were grown under two different culture conditions (1: including all cell types, 2: classic Gary Banker's Protocol (Face-to-face culture of low-density neurons on coverslips and astrocytes on culture dishes). The authors inhibited astrocyte proliferation by adding Ara-C, but this treatment does not completely kill astrocytes, so the possibility that large numbers of astrocytes remain in the culture system cannot be excluded.

In addition, NeurobasalTM is used as the culture medium. This medium contains 0.4 mM of L-serine, a concentration sufficient to provide the extracellular supply of L-serine necessary for the survival and development of cultured neurons. In other words, neurons can take up sufficient amounts of L-serine from the medium.

The above characteristics of culture conditions affect the results of a significant portion of this study.

Major concerns:

1. Does the APL-AIS shown in Figure 1 detect biotinylated AIS protein with the Anti-PHGDH antibody as opposed to the Anti-NFASC antibody? If PHGDH is localized in AIS, it should be detectable by this method as well. At the same time, it is necessary to indicate the ratio of AIS-localized protein in the total protein biotinylated by APL-AIS with the PHGDH antibody.

2. To demonstrate localization of endogenous PHGDH to AIS, it needs to be biochemically demonstrated in ways other than APL-AIS. Can the interaction of AIS protein and endogenous PHGDH be detected by immunoprecipitation or chemical cross-linking from brain tissue and cultured neurons?

3. Figure 4B presents immunocytological staining image data of cultured neurons regarding the localization of PHGDH to AIS. PHGDH signals were detected not only in the vicinity of AIS, but also dispersed throughout the cytoplasm and extracellularly. This image data also has a low resolution. In addition to conventional neuroanatomical findings (Yamasaki et al, J Neurosci, 2001), recent proteomic analysis (Chai et al, Neuron, 2017) confirms that Phgdh is concentrated in astrocytes. Therefore, it cannot be denied that the PHGDH-stained image shown in 4B may detect PHGDH distributed in the astrocytes and their processes lining the MAP-positive neurons. Cultured cells should be stained with AnkG, PHGDH, and an astrocyte-specific marker (Aldh1L1), and three-dimensional signal distribution analysis should be performed to confirm localization in the AIS of cultured neurons. The authors ectopically expressed V5-tagged PHGDH using an AAV vector and show localization to AIS (Fig. 4F), although some V5 signals appears to be also present in extraneuronal processes. e. Forced expression using viral vectors may show artifacts, so more clear analysis results are needed for the distribution of PHGDH endogenously expressed in neurons (including distribution frequencies other than AIS).

4. No quantitative data have been presented on the proportion of neurons with PHGDH distribution in the AIS. Is PHGDH localized to AIS in all neurons? Quantitative analysis results should be shown as to whether the percentage of neurons in which PHGDH is distributed in AIS.

5. The PHGDH staining image of the brain tissue section in Figure S5 is only data from 7 days after birth, the resolution is extremely low, and no quantitative analysis of the distribution has been performed. Therefore, the AIS localization of PHGDH in situ is unclear and unconvincing. Localization of PHGDH in AIS in brain tissue needs to be confirmed by more rigorous methods. Three-dimensional distribution analysis of double immunofluorescent-labeled brain sections at high magnification by confocal laser microscopy, or colocalization analysis of PHGDH with AIS marker protein by immunogold electron microscopy are essential for nailing down of the AIS localization. Also, whether the PHGDH signal in AIS is internal or extracellular in neurons needs to be defined. Furthermore, it is necessary to show whether the distribution of PHGDH in AIS is maintained not only at postnatal day 7 but also at subsequent developmental stages (P14, 21).

6. Central neurons are classified into different types (excitatory output signal cells such as pyramidal neurons, non-pyramidal neurons, excitatory and inhibitory). Regarding the distribution of PHGDH to AIS in brain, it is necessary to identify the types of neurons detected with specific markers and clarify which types of neurons have PHGDH in AIS.

Minor concerns:

1. It is not stated in figure legends which of the two culture conditions was used for all experimental results, except for inhibitor experiments. This statement is required for all presented data, including AIS-proteome analyses.

2. In the two culture conditions, the ratio of astrocytes to neurons determined by cell type marker mRNA/protein expression at each developmental stage used in the experiment (DIV7, 14, 21) need to be shown quantitatively.

II. Is PHGDH-mediated intraneuronal serine synthesis really necessary for the construction and function of AIS?

A conceptual novelty claimed by the authors in this study is that PHGDH activity is required for the structure and function of AIS in neurons. In order to demonstrate this, the authors claim that the enzymatic activity of PHGDH contributes to the formation of AIS, based on the reduction in signals in immunostaining of AIS size and constituent proteins due to treatment with a PHGDH inhibitors (Fig. 5). Similar inhibitor experiments have shown that PHGDH activity also contributes to the generation of action potentials (Fig. 5). However, there are multiple concerns with the interpretation of the results of these experiments.

First, the PHGDH inhibitor NCT503 used in the experiment exhibits an off-target effect and inhibits the metabolism of pyruvate to citrate independently of PHGDH (Arlt et al, J Enzyme Inhib Med Chem, DOI: 10.1080/14756366.2021.1935917). At the same time, it promotes the conversion of pyruvate to malate, and these two changes result in metabolic remodeling of the glucose-TCA cycle. These off-target effects have been reported in experiments with neuroblastoma cells. Moreover, the off-target effect was observed at 0.01 mM, which is lower than the 0.05 mM used by Zhang et al in this submitted study. The concentration used by the authors certainly caused remodeling of the TCA circuit, which likely induced misconfiguration of AIS. The authors also used another PHGDH inhibitor, CBR5884, and showed a similar, somewhat weaker effect, but the off-target effects of this inhibitor were never investigated.

In addition, Neurobasal medium is used in the culture conditions of this study, and this medium contains 0.4 mM L-serine (<https://www.thermofisher.com/jp/ja/home/technical-resources/media-formulation.251.html>). This concentration is higher than the L-serine concentration in blood (0.1-0.2 mM). Therefore, a medium containing L-serine at a high concentration of about 0.4 mM promotes good survival and process outgrowth of hippocampal neurons (Mitoma et al, Neurosci Res, 1998).

So why do PHGDH promote AIS assembly in the presence of extracellular L-serine at higher concentration?

In order to demonstrate that the serine synthesis activity of PHGDH contributes to AIS assembly in neurons, which the author claims, the following experimental verification is necessary.

1. Neuron-specific knockdown (KD) of PHGDH with shRNA expression vector or CRISPR-Cas9 system. This can be done with neurons cultured on coverslips. This analysis can prove that PHGDH actually contributes to the construction of AIS in neurons.

2. Perform similar KD experiments with PSAT1 and PSPH, which are members of the serine synthesis pathway (SSP) in addition to PHGDH. These KD experiments allow us to demonstrate beyond doubt that intraneuronal serine synthesis plays a role in AIS assembly.

3. Apply this KD system to electrophysiological analysis similar to Figure 6 to verify its contribution to AIS functions such as action potential generation.

Reviewer #3 (Remarks to the Author):

Comments to the manuscript by Zhang et al.

(Manuscript ID: #NCOMMS-23-12013)

Title; Spatially Resolved Proteomic Profiling Uncovers Structural and Functional Regulators of the Axon Initial Segment

In this manuscript, the authors have developed an antibody-targeted proximity labeling method APL-AIS to explore the axon initial segment (AIS)-enriched proteins that control AIS structure and function in cultured neurons during development. After fixation and membrane permeabilization of cultured neurons, proteins were biotin-labeled using primary antibodies that recognize NFASC cells localized at the AIS and HRP-conjugated secondary antibodies that recognize them, followed by visualization, purification, and mass spectrometry. According to this approach, the authors concluded that phosphoglycerate dehydrogenase (PHGDH) was localized at AIS in vitro and in vivo and regulated AIS structure and function during neuronal development.

My major concerns are that (1) antibody targeted proximity has already been reported and isn't a novel technique, except that it has been adjusted to neuronal developmental process; (2) AIS proteome analysis has been performed in vitro and in vivo in recent years, and there is no significant improvement in information compared to those previous AIS proteome analyses; (3) Antibodies recognizing the

extracellular region of NFASC were used, but concerns remain in terms of the specificity and efficacy of this approach because the AIS proteome in the manuscript contains many intracellular proteins, and (4) as a result of only using inhibitors, no molecular mechanisms and biological significance have been found to explain the formation and function of AIS.

Major comments

1. On P4, lines 72-75, it would be better to discuss and evaluate the advantages of APL-AIS over previous antibody-dependent BioIDs (Bar et al., 2018; Dopie et al., 2020; Li et al., 2022). Furthermore, the authors have previously reported the AIS proteome in cultured neurons using NF186-BirA (Hamdan et al., 2020), which is the same target AIS protein as this manuscript, and another group has shown endogenous AIS proteome using CRISPR-based endogenous Anks1b-TurboID in the cortex (Gao et al., 2022). If you compare the APL-AIS proteome to these previous proximity labeling-based AIS proteomes, what advantages or new information would be beneficial to the widest possible readers?
2. As shown in Fig. 2-3, a ratio should be given of how many of the total proteins are known AIS proteins and which proteins (membrane proteins, receptors, channels, adhesion molecules, cytosolic proteins, secreted molecules) are included. The AIS proteome data set should also be compared with the BioID probes and CRISPR-based BioIDs in the previous AIS proteome papers (Hamdan et al., 2020; Gao et al., 2022).
3. In Fig. 1C and Fig. S3, in the case of biotin-labeled proteins, are NFASC, NeuN, MAP2, and SIM1312 detected as positive controls? It is important to evaluate APL-AIS before performing mass spectrometry.
4. Since most of the proteins in Fig. 1E have a ratio less than 1, would it be reasonable to assume many of the proteins identified by mass spectrometry are non-specific or other proteins rather than specific AIS proteins? Protein types (known AIS proteins, receptors, channels, adhesion molecules, cytosolic proteins, secretory molecules, cytosolic proteins) should be classified and their ratios should be shown.
5. In Fig. 1-3, there is a major concern that anti-NFASC recognizes the extracellular region of NFASC, but why have so many intracellular proteins been found? In biotinylated labeled peptides, are there more extracellular regions?
6. Its major advantage is its high time resolution since Anti-NFASC recognizes extracellular regions and is biotin-labeled within 1 minute, allowing mass spectrometry to be performed even in living cells. In my opinion, the findings and manuscript will be extremely valuable if the authors are able to demonstrate the APL-AIS proteome from living cells.
7. Fig. 3 needs GO term analysis to see the protein components identified in DIV7 and DIV21, and also should show what proteins are included (ratio) in the same way as above (Major comments #2 and #4).
8. Fig. 3H-N, p10 line 233-248, the mass spectrometry data set evaluation experiment is required. Specifically, immunoblotting should be performed on these candidate molecules and the actual expression variation of these proteins should be examined during neurodevelopment.
9. In Fig. 4F, localization analysis with CRISPR is very effective and high-throughput for determining unknown proteins' locations. By performing localization analyses on other novel AIS candidate

molecules, instead of only PHGDH and SCRIB, the author should validate the results of the APL-AIS data set.

10. In Fig. 5-Fig. 6, it is not clear how AIS functions by PHGDH because inhibitor-only experiments are not sufficient. It is imperative that authors demonstrate PHGDH-induced AIS formation through knockdown experiments. In vivo experiments should also investigate the effect of PHGDH on AIS formation and function. Additionally, the molecular mechanism of PHGDH regulation of AIS formation and function should be demonstrated: does it bind to NFASC, beta-spectrin, AnkG, or TRIM46?

Minor comments

1. In P5, lines 107-110, the reactivity of the antibody should be shown as a supplemental figure.
2. In Fig. 1B, if the authors show the biotin labeling time, it would be helpful for readers.
3. There is confusion in the criteria shown in Fig. 2C, which says Ratio, but is it subtracting instead of dividing?
4. Table should show raw mass spectrometry data.
5. The in vivo localization analysis is very important; Fig. S5C should be moved to the main figure.

References

Bar, D.Z., Atkatsch, K., Tavares, U., Erdos, M.R., Gruenbaum, Y., and Collins, F.S. (2018). Biotinylation by antibody recognition—a method for proximity labeling. *Nat Methods* 15, 127-133.

Dopie, J., Sweredoski, M.J., Moradian, A., and Belmont, A.S. (2020). Tyramide signal amplification mass spectrometry (TSA-MS) ratio identifies nuclear speckle proteins. *The Journal of cell biology* 219.

Hamdan, H., Lim, B.C., Torii, T., Joshi, A., Konning, M., Smith, C., Palmer, D.J., Ng, P., Leterrier, C., Oses-Prieto, J.A., et al. (2020). Mapping axon initial segment structure and function by multiplexed proximity biotinylation. *Nature communications* 11, 100.

Li, X., Zhou, J., Zhao, W., Wen, Q., Wang, W., Peng, H., Gao, Y., Bouchonville, K.J., Offer, S.M., Chan, K., et al. (2022). Defining Proximity Proteome of Histone Modifications by Antibody-mediated Protein A-APEX2 Labeling. *Genomics Proteomics Bioinformatics* 20, 87-100.

Gao Y., Trn M., Shonai D., Zhao J., Soderblom EJ., Garcia-Moreno SA., Gersbach CA., Wetsel WC., Dawson G., Velmeshev D., Jiang Y., Sloofman L., Buxbaum JD., Soderling SH. (2022). Chemico-genetic Analysis of Native Autism Proteomes Reveals Shared Biology Predictive of Functional Modifiers. *bioRxiv*. DOI: 10.1101/2022.10.06.511211

RESPONSE TO REVIEWER COMMENTS

This revision has 4 completely new figures (and one supplemental figure), all focused on SCRIB. In this revision we do not respond to concerns regarding PHGDH since we removed that work and no longer have confidence in the interpretation of the prior results.

Reviewer #1:

This paper uses a novel proteomic approach to characterise the AIS proteome in cultured neurons through development, and to identify some potentially important new AIS-localised components. This is a valuable dataset which appears extremely carefully produced – the comparison of AIS-localised vs soma/dendrite/axon-localised proteins is especially powerful. The authors also present data from pharmacological manipulations suggesting that PHGDH activity is important for the structural and functional integrity of the mature AIS. I have a few relatively minor concerns regarding these latter experiments, but overall the standard of data collection and presentation is high and the results largely convincing.

We thank the reviewer for their encouraging comments – especially as they relate to the quality of our mass spectrometry results.

Questions 1-5 dealt with PHGDH results which have been removed and are no longer applicable to the revision.

6) Line 100: the authors performed a detailed optimisation of all the parameters in the proximity labelling system. Both the optimal numbers for H₂O₂ concentration (100 μM) and the time (1 minute) are mentioned in the text, but not the chosen concentrations for primary and secondary antibodies. What does ‘proper dilution of antibodies’ mean? What was prioritised for these? Was it a combination between an acceptable ratio of biotin detected and specificity? Or only specificity?

We chose maximum values for the AIS/Soma biotin ratio. When the ratios were comparable between conditions, we chose parameters for generating higher AIS biotinylation such as when performing HRP conjugated secondary antibody optimization. We now specify this in the results.

7) Figure legend Figure S1: please, specify how many cells were analysed for the graphs. Is each dot one cell?

Each dot represents one cell. This is stated in the figure legends.

8) Line 137 and Figure 1 legend: add the stage of the neurons (‘DIV14’) to the figure legend in D to keep the same style within the legend annotation.

Done

9) Figure 1C: please, add an indicator for molecular weight in the blots.

Done

10) Figure S3: please, add an indicator for molecular weight in the blots.

Done

11) N/A

12) N/A

13) *Figure legend of figure S1: in line with the rest of figure legends, please include the chosen statistical test at the bottom of the legend.*

Done

14) *In Methods, the section 'Detergent extraction' (line 794) could go to 'Immunocytochemistry and immunohistochemistry' instead (line 746).*

Done

15) *Methods, line 607: please, provide the details about the protein loading buffer composition.*

Done

16) *More details about how the periodicity was measured in figure 4 are needed in the methods section.*

Done

Reviewer #2:

...However, there are concerns regarding methodology and data interpretation that cannot be ignored and should be resolved. Specifically, concerns need to be resolved regarding the two main novel findings the authors claim, the localization of PHGDH to the neuronal AIS and the contribution of serine synthesis by PHGDH in the assembly and function of the AIS.

We thank the reviewer for their critical comments regarding PHGDH at the AIS and its functions there. The comments prompted us to test CRISPR-mediated knockdown of PHGDH. However, we were unable to remove the AIS immunostaining. We thought our sgRNAs were simply not working. Therefore, we imported the floxed *Phgdh* mice (from Japan) and begin working on them to answer all of these questions. However, after importing and crossing the mouse lines to generate neuron specific loss of PHGDH, we were unable to confirm loss of the AIS labeling. This immediately caused us to re-evaluate all our conclusions regarding PHGDH. Unfortunately, we are no longer confident in our interpretation and results, therefore we removed all PHGDH experiments from this manuscript. We are very grateful to the reviewer for demanding that we re-examine our conclusions more rigorously. We agree with the reviewer that some of our prior observations on the effect of PHGDH inhibitors could be due to effects on astrocytes in the cultures. Nevertheless, due to the extensive revision and removal of all PHGDH results, all concerns raised by this reviewer are no longer applicable.

Reviewer #3:

My major concerns are that (1) antibody targeted proximity has already been reported and isn't a novel technique, except that it has been adjusted to neuronal developmental process;

Antibody directed proximity biotinylation has been performed by others, but this is the first time it has been applied to the entire AIS. While this work was underway, one of us (Rasband lab) independently (and without the knowledge of the other) performed antibody directed surface proximity biotinylation (this paper was recently accepted for publication and is in press). This work is different as it was applied to the entire AIS after fixation and permeabilization. In addition, the use of reference (somatodendritic and axonal) compartments to further restrict analyses to only AIS enriched proteomes is a significant technical difference and improvement.

(2) AIS proteome analysis has been performed in vitro and in vivo in recent years, and there is no significant improvement in information compared to those previous AIS proteome analyses;

Respectfully, we disagree with this statement. Each method has its own limitations – for example, the BioID approach (Hamdan et al., 2020) has a limited range and yielded almost no membrane proteins, but several new cytoplasmic AIS proteins. Nevertheless, the utility of the approach is in identification of new candidates and confirmation of previously reported AIS proteins. With these metrics, this data set is the most robust and comprehensive of any to be reported to date.

(3) Antibodies recognizing the extracellular region of NFASC were used, but concerns remain in terms of the specificity and efficacy of this approach because the AIS proteome in the manuscript contains many intracellular proteins.

Perhaps the reviewer has misunderstood – the experiments designed here were intended to identify both cell surface proteins and intracellular proteins. We used detergents in these experiments to extend the labeling to intracellular proteins. This is in contrast to Yuki et al., (2023 bioRxiv and in press), where no detergents were used to restrict labeling to the cell surface.

(4) as a result of only using inhibitors, no molecular mechanisms and biological significance have been found to explain the formation and function of AIS.

We removed the inhibitor experiments and all PHGDH experiments. We now focus on SCRIB and show mechanistically that SCRIB is a novel AIS protein that is recruited to the AIS through direct binding to AnkG. We are working on additional mechanistic studies of SCRIB to determine what SCRIB does at the AIS, but we believe this is beyond the scope of this proximity biotinylation study.

Major comments

1. On P4, lines 72-75, it would be better to discuss and evaluate the advantages of APL-AIS over previous antibody-dependent BioIDs (Bar et al., 2018; Dopie et al., 2020; Li et al., 2022). Furthermore, the authors have previously reported the AIS proteome in cultured neurons using NF186-BirA (Hamdan et al., 2020), which is the same target AIS protein as this manuscript, and another group has shown endogenous AIS proteome using CRISPR-based endogenous Anks1b-TurboID in the cortex (Gao et al., 2022). If you

compare the APL-AIS proteome to these previous proximity labeling-based AIS proteomes, what advantages or new information would be beneficial to the widest possible readers?

The immunoproximity labeling (IPL-AIS) approach is not necessarily better than other strategies; it is a different strategy with its own advantages and weaknesses. It depends on the question and how it is applied. The major difference between this work and Hamdan et al. (2020) is that we used an antibody-based approach that does not rely on transfection and over-expression of a Bio-ID-containing fusion protein. The antibody-based approach gives a much larger labeling radius of ~250 nm in fixed samples, while the BioID approach is limited to ~10 nm. Neither is necessarily 'better' – it depends on the questions being asked. We describe some of the differences in the approach in the introduction. As for the CRISPR-based endogenous TurboID, this is also a very good strategy, but suffers from several issues: 1) low transduction efficiency with the large TurboID, 2) addition of the TurboID occurs in coding sequence and therefore disrupts the remaining part of the protein, and 3) is limited spatially, and 4) cannot be used to limit the biotinylation to the extracellular surface proteins only (although that wasn't the purpose of either study reported here. To emphasize the point, although the paper of Gao et al. is remarkable in its scope and ambition, at no point do the authors actually demonstrate the efficacy of the TurboID knock-in to the endogenous targeted proteins. This is shown with smFP, but not for TurboID and the authors do not show biotinylation of the target proteins restricted to the appropriate domains. The proteomics also do not match the known proteome for the AIS, instead also identifying many proteins that are NOT in the AIS. Our point is not to criticize that paper, but to only illustrate that each approach has its own advantages and limitations. Each approach must be viewed in the context of the proposed experiments and each will yield different information. Thus, there is not a 'one-size-fits-all' approach to proteomics and proximity biotinylation of AIS proteins.

2. As shown in Fig. 2-3, a ratio should be given of how many of the total proteins are known AIS proteins and which proteins (membrane proteins, receptors, channels, adhesion molecules, cytosolic proteins, secreted molecules) are included. The AIS proteome data set should also be compared with the BioID probes and CRISPR-based BioIDs in the previous AIS proteome papers (Hamdan et al., 2020; Gao et al., 2022).

Previously reported AIS proteins are included in the figures as red dots and can be found in the supplementary tables. All prior identifications of AIS proteins (many based on Hamdan et al., 2020) are included in the results and presentation – see Figures 1-4. We have not included the results compared to Gao et al. as that is a preprint and has not been peer reviewed. We chose not to give a percentage of known AIS to those not previously reported since this may give a false impression of knowledge about the nature of a protein. We do not imply that all proteins identified here are AIS proteins.

3. In Fig. 1C and Fig. S3, in the case of biotin-labeled proteins, are NFASC, NeuN, MAP2, and SIM1312 detected as positive controls? It is important to evaluate APL-AIS before performing mass spectrometry.

Yes, these are given in the supplemental tables.

4. Since most of the proteins in Fig. 1E have a ratio less than 1, would it be reasonable to assume many of the proteins identified by mass spectrometry are non-specific or other proteins rather than specific AIS

proteins? Protein types (known AIS proteins, receptors, channels, adhesion molecules, cytosolic proteins, secretory molecules, cytosolic proteins) should be classified and their ratios should be shown.

We used log₂ fold change to analyze 568 proteins in common between soma and somatodendritic domains in the pilot experiment. Proteins with a ratio less than 1 may be due to the protein being found in both domains. This does not mean the protein is NOT an AIS protein, only that it is not exclusively or highly enriched in the AIS. For example, α 2 spectrin is well recognized to be important for AIS assembly and maintenance (Huang et al., J Neurosci 2017). However, it is not exclusively found in the AIS (see Hamdan et al., Nat Comm 2020). Thus, additional validation experiments must be performed to exclude or include any given protein as a bona fide AIS protein. This is exactly what we have done for Scrib.

5. In Fig. 1-3, there is a major concern that anti-NFASC recognizes the extracellular region of NFASC, but why have so many intracellular proteins been found? In biotinylated labeled peptides, are there more extracellular regions?

We remind the reviewer that the experiments were designed to biotinylate both extracellular, membrane, and cytoplasmic proteins at the AIS. The neurons were fixed and membranes were permeabilized using detergent. The range of biotinylation with this method is ~250 nm. So there is no concern and the identification of cytoplasmic proteins is absolutely expected – and wanted. This is in contrast to Ogawa et al., (2023) where the biotinylation was performed on live, unfixed neurons, thereby limiting the labeling to the cell surface.

6. Its major advantage is its high time resolution since Anti-NFASC recognizes extracellular regions and is biotin-labeled within 1 minute, allowing mass spectrometry to be performed even in living cells. In my opinion, the findings and manuscript will be extremely valuable if the authors are able to demonstrate the APL-AIS proteome from living cells.

We agree that could be an interesting experiment. However, we did not optimize the IPL-AIS method for live cells, but rather for fixed, permeabilized cells. The purpose of our experiments was to define the full AIS proteome which had to be done not on live cells, but permeabilized and fixed cells. Respectfully, we have still identified new candidates and we consider this to be extremely valuable. Among the first ones we tested we found SCRIB is highly enriched at the AIS and is specific to the AIS. Current studies are underway to evaluate the many other potential AIS proteins.

7. Fig. 3 needs GO term analysis to see the protein components identified in DIV7 and DIV21, and also should show what proteins are included (ratio) in the same way as above (Major comments #2 and #4).

In our experiments we carefully ranked putative AIS proteins. Nevertheless, we do not think all candidates are restricted to the AIS and it is difficult to perform a cut-off for GO term analysis due to the current reference dataset of GeneOntology consisting of less than 30 proteins. To avoid confusion, we prefer not to show GO term analysis. Ratios for those proteins were provided in supplementary tables.

8. Fig. 3H-N, p10 line 233-248, the mass spectrometry data set evaluation experiment is required. Specifically, immunoblotting should be performed on these candidate molecules and the actual expression variation of these proteins should be examined during neurodevelopment.

The results presented by mass spectrometry may not match immunoblotting results. Since the biotinylation occurs on tyrosine residues, different proteins may have different numbers of Tyrosines and accessibility of Tyrosines to biotinylation. Thus, immunoblotting is not necessarily a good way to validate or confirm the changes reported here by mass spectrometry. Nevertheless, the trends and many of the results fit well with previously reported changes in expression levels of AIS proteins. See for example, Ank3 (Galiano et al., Cell 2012), Nav channels (Yang et al., JCB 2007), Nfasc (Ogawa et al., Nat Comm, in press).

9. In Fig. 4F, localization analysis with CRISPR is very effective and high-throughput for determining unknown proteins' locations. By performing localization analyses on other novel AIS candidate molecules, instead of only PHGDH and SCRIB, the author should validate the results of the APL-AIS data set.

We agree with this reviewer's suggestion that CRISPR knock-in will be useful to validate and test our candidates. Indeed, we are doing exactly this. For this paper we report the top 3 candidates: Wdr7, Scrib, and Wdr47. We believe it is beyond the scope of this paper to generate sgRNA and AAVs to validate all of our top candidates.

10. N/A

Minor comments

1. In P5, lines 107-110, the reactivity of the antibody should be shown as a supplemental figure.

The reactivity of the antibody is shown in Fig. 1b.

2. In Fig. 1B, if the authors show the biotin labeling time, it would be helpful for readers.

In this study we used one minute of biotin labeling reaction in all experiments – this has been added to the figure legend in Fig. 1B.

3. There is confusion in the criteria shown in Fig. 2C, which says Ratio, but is it subtracting instead of dividing?

We used \log_2 fold change for the calculation. Please note: $\log_2(A/B) = \log_2A - \log_2B$.

4. Table should show raw mass spectrometry data.

Raw mass spectrometry data are included in ProteomeXchange with identifier (PXD045921). Output data from MaxQuant software (version 1.6.10.43) and ratios are provided in supplementary tables 1-3.

5. N/A

REVIEWERS' COMMENTS

Reviewer #1 (Remarks to the Author):

This revision addresses all of my relevant previous concerns. In addition, the new data on SCRIB are solid and support the claims made in the manuscript. This more than compensates for the removal of the PHGDH data - which I fully support - and makes this paper a very nice demonstration of how a well-performed screen can identify new and important AIS-localised proteins.

Reviewer #3 (Remarks to the Author):

The authors have addressed my primary concerns by incorporating new results and conducting further data analysis, which enhances the quality of the paper compared to the previous version. However, I have some comments regarding the IPL-AIS method and the physiological role of ANKG-SCRIB.

1. The revised manuscript presents new data indicating that SCRIB is recruited to the AIS through its interaction with ANKG. However, the role of SCRIB in the AIS remains unclear. The authors should rigorously investigate its physiological significance in this context.
2. The authors have incorporated the statement: 'The neurons were fixed and the membranes were permeabilized using detergent. The expected range of biotinylation with this method is approximately 250 nm.' It is vital to include these sentences in the manuscript and expand upon them in the discussion section. This is crucial because it highlights that the proximity labeling of IPL-AIS can label both cell surface molecules and intracellular proteins. This unique capability positions this study at an advantage over similar AIS proteomes that have been previously reported (e.g., Ogawa et al., in press; Hamda et al., 2020).
3. I suggest moving the localization data for endogenous SCRIB, currently in Figure 6e, to Figure 5. Relocating this data would reinforce the results from the localization analysis obtained by knockin.

RESPONSE TO REVIEWERS

Reviewer #1 (Remarks to the Author):

This revision addresses all of my relevant previous concerns. In addition, the new data on SCRIB are solid and support the claims made in the manuscript. This more than compensates for the removal of the PHGDH data - which I fully support - and makes this paper a very nice demonstration of how a well-performed screen can identify new and important AIS-localised proteins.

We thank the reviewer for their time and effort reviewing this paper.

Reviewer #3 (Remarks to the Author):

The authors have addressed my primary concerns by incorporating new results and conducting further data analysis, which enhances the quality of the paper compared to the previous version. However, I have some comments regarding the IPL-AIS method and the physiological role of ANKG-SCRIB.

1. The revised manuscript presents new data indicating that SCRIB is recruited to the AIS through its interaction with ANKG. However, the role of SCRIB in the AIS remains unclear. The authors should rigorously investigate its physiological significance in this context.

We completely agree that investigating the physiological role of SCRIB is important and we assure the reviewer we are doing exactly this. However, we believe this is beyond the scope of the present manuscript which is focused on using antibody directed proximity biotinylation to identify new AIS proteins.

2. The authors have incorporated the statement: 'The neurons were fixed and the membranes were permeabilized using detergent. The expected range of biotinylation with this method is approximately 250 nm.' It is vital to include these sentences in the manuscript and expand upon them in the discussion section. This is crucial because it highlights that the proximity labeling of IPL-AIS can label both cell surface molecules and intracellular proteins. This unique capability positions this study at an advantage over similar AIS proteomes that have been previously reported (e.g., Ogawa et al., in press; Hamda et al., 2020).

We added this information and have emphasized it in the introduction and discussion as requested.

3. I suggest moving the localization data for endogenous SCRIB, currently in Figure 6e, to Figure 5. Relocating this data would reinforce the results from the localization analysis obtained by knockin.

We respectfully prefer to keep the current arrangement. Figure 5 shows data from endogenous genome labeling to tag endogenous SCRIB *in vitro* and *in vivo*, while Figure 6 shows results for SCRIB labeling *in vitro* and *in vivo* from immunostaining. The two results are consistent but rely on entirely different experimental approaches.